# Driver gene combinations dictate cutaneous squamous cell carcinoma disease continuum progression

Peter Bailey [1,2,3] ✉, Rachel A. Ridgway [4], Patrizia Cammareri[4,12],
Mairi Treanor-Taylor [4,5], Ulla-Maja Bailey [4], Christina Schoenherr [4],
Max Bone [1,4], Daniel Schreyer [1], Karin Purdie[6], Jason Thomson[6,7],
William Rickaby[8], Rene Jackstadt[4,13], Andrew D. Campbell [4],
Emmanouil Dimonitsas[9], Alexander J. Stratigos[9], Sarah T. Arron [10],
Jun Wang [6], Karen Blyth [1,4], Charlotte M. Proby [11], Catherine A. Harwood[6,7],
Owen J. Sansom [1,4], Irene M. Leigh [6] ✉ & Gareth J. Inman [1,4] ✉

The molecular basis of disease progression from UV-induced precancerous actinic keratosis (AK) to malignant invasive cutaneous squamous cell carcinoma (cSCC) and potentially lethal metastatic disease remains unclear. DNA sequencing studies have revealed a massive mutational burden but have yet to illuminate mechanisms of disease progression. Here we perform RNAseq transcriptomic profiling of 110 patient samples representing normal sun-exposed skin, AK, primary and metastatic cSCC and reveal a disease continuum from a differentiated to a progenitor-like state. This is accompanied by the orchestrated suppression of master regulators of epidermal differentiation, dynamic modulation of the epidermal differentiation complex, remodelling of the immune landscape and an increase in the preponderance of tumour specific keratinocytes. Comparative systems analysis of human cSCC coupled with the generation of genetically engineered murine models reveal that combinatorial sequential inactivation of the tumour suppressor genes *Tgfbr2*, *Trp53*, and *Notch1* coupled with activation of Ras signalling progressively drives cSCC progression along a differentiated to progenitor axis. Taken together we provide a comprehensive map of the cSCC disease continuum and reveal potentially actionable events that promote and accompany disease progression.

Cutaneous squamous cell carcinoma (cSCC) represents a significant and rising global health burden[1] and one of the commonest malignancies with metastatic potential[2,3]. Histological examination is used to assign clinical definitions of AK premalignant disease or invasive cSCC. Clinicopathological staging is an important determinant of clinical pathway decision making, although current staging systems have limited prognostic utility[4–6]. Surgical excision or radiotherapy are treatments of choice for primary cSCC and adjuvant radiotherapy is used in certain "high-risk" primary tumours[7]. In locally advanced and metastatic disease not suitable for surgery or radiotherapy, responses to chemotherapy and/or Epidermal Growth factor Receptor (EGFR) inhibitors are limited and although anti-Programmed Cell Death 1 (PD1) checkpoint immunotherapy is now considered first line, 50% of individuals fail to respond[7] highlighting the need for better treatment modalities.

The genetic architecture of both premalignant AK and invasive cSCC is complex, with both displaying high mutational burdens and mutational heterogeneity, copy number changes and shared genetic alterations in some common driver genes[8–11]. Mutations of these driver genes are also observable in clinically "normal" skin from sun exposed and other body sites[12,13] further complicating the potential of genotype treatment selection decisions. Current estimates indicate that fewer than 0.1% of individual AKs will progress to cSCC and that 3–5% of primary cSCC have the potential to progress to life threatening metastatic disease[14]. Previous cross species studies integrating gene expression and genetic analysis have revealed potential mechanisms of cSCC disease progression[8] but there remains a clear and urgent need for a deeper understanding of the biological processes that underpin disease progression not only for improved risk-stratification to support rational deployment of treatment but also to identify new possible therapeutic strategies.

Here we perform RNAseq on 110 samples spanning the spectrum of human cSCC disease from normal sun exposed skin, AK, primary cSCC and metastasis and demonstrate that disease progression can be represented as a continuum from a differentiated to a progenitor like state. We catalogue the changes in genes, biological pathways, processes and cell types through this continuum. Utilising genetically engineered mouse models we recapitulate these events and demonstrate that driver gene combinations dictate disease progression.

## Results

### Transcriptomic analysis defines gene expression profiles associated with actinic keratosis (AK) and primary squamous cell carcinoma (SCC)

To characterise the molecular mechanisms underpinning cSCC progression we performed bulk RNAseq analysis on 110 treatment-naïve patient samples representing normal sun exposed skin/peri-lesional skin ($n = 26$), AK ($n = 14$), primary ($n = 66$) and metastatic cSCC ($n = 4$) (Supplementary Data File 1). Principal Component Analysis (PCA) demonstrated that normal, AK and primary cSCC diverge along dimension 1 (Supplementary Fig. 1a). To characterise changes in gene expression between clinical stage we performed differential gene expression analysis comparing normal versus AK, AK versus primary cSCC and normal versus primary cSCC (Supplementary Data File 2). This analysis identified unique subsets of genes defining each clinical stage with the normal versus primary cSCC comparison generating the largest number of differentially expressed genes. Subsequent gene set enrichment analysis (GSEA) identified several molecular pathways and/or processes significantly enriched in either normal, AK or primary cSCC samples (Supplementary Fig. 1b–d). These included marked modulation of metabolic, proliferation and immune signalling processes (Supplementary Data File 2). Importantly enrichment in cell cycle related processes were observed in both the normal to AK and normal to primary tumour comparisons indicating the hyperproliferative nature of both pre-malignant and malignant disease (Supplementary Fig. 1b, d). Extracellular matrix processes were altered in normal-AK and AK-primary tumour transitions and cytokine, chemokine and IFN signalling pathways were enriched in primary tumour samples compared to both normal and AK samples indicating the potential importance of these events in primary tumour formation (Supplementary Fig. 1c, d). Our sample set contained several samples that were collected from the same patients including 25 matched pairs of normal skin and primary tumours (Supplementary Fig. 2a). We performed matched sample differential gene expression analysis on these samples (Supplementary Data File 3) and compared this to the differential gene expression analysis of the whole normal and primary tumour data set (normal skin $n = 26$, primary tumour $n = 66$). There was considerable overlap of DEGs between these two analyses with 1357 downregulated and 915 upregulated genes ($p$adj <0.05, log2FC >1) in primary tumours compared to normal samples in both

analyses (Supplementary Fig. 2b). The whole sample set analysis revealed an additional 343 downregulated and 567 upregulated genes whereas the matched analysis revealed an additional 380 downregulated and 39 upregulated genes with the additional identification of neuronal and muscle contraction processes identified in GSEA analysis of the matched samples and IFNγ and chemokine signalling additionally identified in the whole sample set analysis (Supplementary Fig. 1d, 2c) highlighting the strengths of both approaches to reveal potentially important pathways and processes in disease progression.

### A cSCC disease progression continuum is associated with the orchestrated suppression of epidermal differentiation and the induction of progenitor-like gene expression

Unsupervised hierarchical clustering and tSNE analysis of our RNaseq samples revealed that samples clustered into two main distinct clusters designated Class 1 and Class 2 (Supplementary Fig. 3a, b). Class 1 samples comprised predominantly normal skin and AK samples whereas Class 2 samples comprised predominantly primary SCC and metastasis samples. GSEA analysis revealed changes in metabolic and immune processes with epidermis development and keratinization highly enriched in Class 1 samples (Supplementary Fig. 3c, Supplementary Data File 4). The epidermis is a stratified self-renewing epithelial tissue that acts as an important outer-barrier to both repel foreign insults and maintain organismal homeostasis[15] and comprises distinct layers of keratinocytes that represent a continuum of epidermal differentiation. The keratinocytes of the epidermis arise from resident stem cells located within the basal cell layer. Upon activation, these stem cells exit the cell cycle and translocate into the supra-basal compartment where they undergo progressive stages of differentiation. Ultimately, keratinocytes terminally differentiate to form enucleated lipid-embedded corneocytes that finally undergo cornification to form the stratum corneum or outer skin.

The transcriptional changes underpinning human epidermal differentiation have been mapped in fine detail. Specific sets of genes defining progenitor cell populations, early differentiation and late (terminal) differentiation (Fig. 1a) have been experimentally validated using organotypic models[16]. We used GSEA to determine if the orchestrated loss of epidermal differentiation observed in cSCC is associated with the dysregulated expression of these specific gene sets (Supplementary Data File 5). This demonstrated that normal skin and AK are significantly enriched for genes associated with late epidermal differentiation, whereas primary and metastatic SCC are significantly enriched for genes associated with skin progenitor cells (Fig. 1b). Similarly, we observed significant differences in late differentiation, early differentiation and progenitor signature scores between Class 1 and Class 2 samples (Supplementary Fig. 3d). Our dataset includes samples from immunocompetent and immunosuppressed patients, but we observed no significant differences in early differentiation and progenitor signature scores between these groups and only a modest enrichment of late differentiation signature scores in immunosuppressed patients (Supplementary Fig. 3d).

The clear delineation of normal and AK from primary and metastatic cSCC using the late epidermal and progenitor-like signatures suggested that cSCC patient samples may represent a continuum of epidermal de-differentiation. In keeping with this notion, cSCC could be stratified into two broad patient groups representing either a "Differentiated" or "Progenitor-like" state (Fig. 1c). Importantly, using a Differentiation-versus-Progenitor (DvP) signature score we reveal that cSCC disease progression can be represented as a disease continuum from a differentiated to a progenitor like state (Fig. 1c, Supplementary Data File 5). Poorly differentiated (PD) primary cSCC tumours are associated with worse prognosis compared to their well differentiated (WD) counterparts[6]. Moderately (MD) - PD and PD samples were significantly associated with the progenitor-like state (Kruskal-Wallis, $p = 0.031$) indicating that

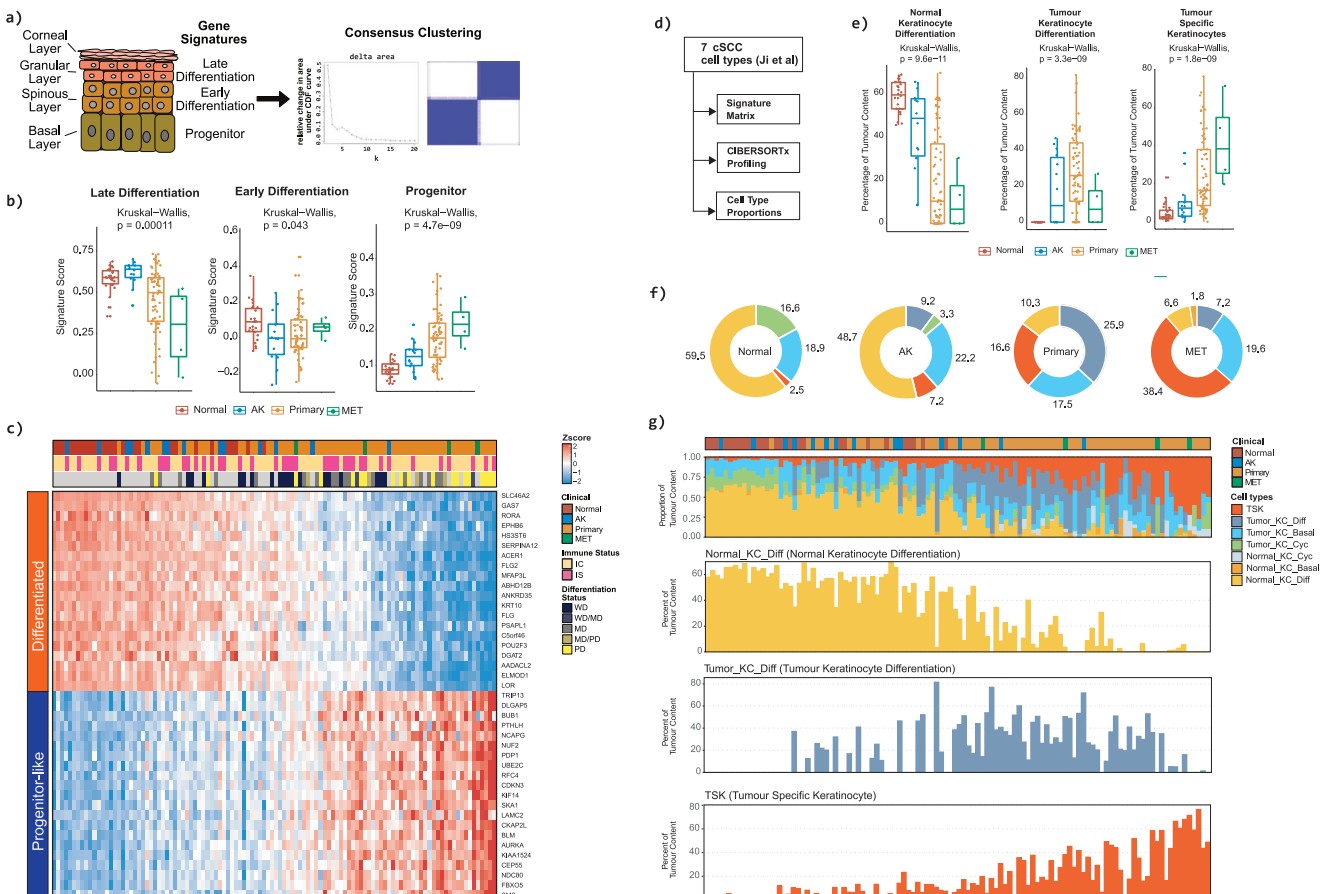

**Fig. 1 | cSCC progression is associated with the orchestrated suppression of epidermal differentiation and the induction of progenitor-like gene expression. a** Diagram of the stratified layers of the skin epithelium and their associated gene signatures (left panel). Consensus clustering of 110 human human samples profiled by RNAseq identifies two classes of samples. **b** Boxplots demonstrating the enrichment of Late Epidermal Differentiation, Early Epidermal Differentiation and Progenitor gene signatures in normal (red, *n* = 26), AK (blue, *n* = 14), primary (orange, *n* = 66) and metastatic cSCC (MET, green, *n* = 4). Boxplots are annotated by a Kruskall-Wallis P value with P values <= 0.05 indicating a significant difference between clinical designations. **c** Heatmap showing the expression of genes associated with late epidermal differentiation and a Progenitor-like state across a spectrum of cSCC clinical designations. The genes shown in the heatmap represent a Differentiation-Progenitor-like (DvP) signature which has been used to order samples along an axis of late epidermal differentiation to progenitor-like gene expression (DvP signature score). Patient Immune status (IC, immunocompetent,

IS, immunosuppressed) and differentiation status of the primary tumours (WD, well differentiated; MD, moderately differentiated; PD, poorly differentiated) are indicated. **d** Schematic diagram showing the CIBERSORTx workflow, which was used to estimate the proportion of defined single cell populations resident in bulk tumour SCC samples. **e** Boxplots showing the estimated percentage of defined single cell populations representing Normal Keratinocytes Differentiated, Tumour Keratinocytes Differentiated and Tumour Specific Keratinocyte cells in Normal (red), AK (blue), Primary (orange) and MET (green) bulk cSCC samples. Boxplots are annotated by a Kruskal-Wallis *P* value with *P* values <= 0.05 indicating a significant difference between clinical designations. **f** Donut charts showing the percent tumour enrichment of defined single cell populations in bulk cSCC samples stratified by clinical designation. **g** Bar charts showing the enrichment of defined single cell populations in bulk cSCC samples ordered according to the DP signature score (KC, keratinocyte; Diff, differentiated; Cyc, cycling). Source data for b and e are provided in the Source Data file.

progenitor score analysis may have primary tumour classification utility (Supplementary Fig. 4a–c). Most progenitor-like samples (DvP quartile 1) had greater tumour depth (Kruskal Wallis, p = 0.02, Supplementary Fig. 4d) and tumour diameter (Kruskal Wallis, *p* = 0.0031, Supplementary Fig. 4e) but DvP score did not significantly associate with invasion status, patient age at time of sampling or sex (Supplementary Fig. 4f–h). The utility of representing cSCC disease progression as a continuum is also exemplified by considering our matched samples of primary tumours and normal skin which are distributed across the DvP axis (Supplementary Fig. 5a) and despite having overall significant differences in DvP signature score and expression of LOR as an example signature gene (Supplementary Fig. 5b, c) individual pairs of samples may show similar signature scores or LOR expression.

Recently, Ji et al. have identified seven distinct single-cell keratinocyte populations resident in normal skin and cSCC including a unique population of tumour specific keratinocytes (TSK)[17]. Using

Signature matrix and CIBERSORTx[18] analysis (Fig. 1d) we calculated the proportions of these seven keratinocyte populations in our sample set (Supplementary Data File 6). During progression through normal-AK-SCC-Metastasis, we found a significant reduction in the proportion of Normal Keratinocyte Differentiated cells (NKD) and an increase in the proportion of TSKs (Fig. 1e, f) and the appearance of a Tumour Keratinocyte Differentiated population (TKD) at the AK stage (Fig. 1e, f). We also uncovered profound changes in keratinocyte populations during progression along the DvP continuum with progenitor like samples containing a high proportion of TSKs (20–80%) which progressively increase with progenitor like state (Fig. 1g). Interestingly we observed a modulation of the tumour keratinocyte cycling population across the continuum, the appearance of the normal keratinocyte cycling population in progenitor like samples and no notable changes in basal keratinocyte populations (Supplementary Fig. 6) and these observations warrant further investigation.

**cSCC progression is associated with the orchestrated gain and loss of key molecular pathways and/or processes associated with epidermal differentiation, immune signalling, metabolism and progenitor-like cell states**

To decipher the molecular pathways and/or processes underpinning SCC progression we performed k-means clustering using the 2000 most significantly differentially expressed genes between clinical designations. This analysis identified 15 clusters of co-ordinately expressed genes (Supplementary Data File 7). Nine of these clusters exhibited significant correlation with the DvP score (Fig. 2a). Gene enrichment analysis demonstrated that the progression of SCC involves the orchestrated suppression of epidermal differentiation pathways (clusters 8, 3, 11, 9 and 2) and induction of gene programmes associated with cell proliferation, cell-cell communication, MET and PDGF signalling pathways and immunomodulation (clusters 10, 1, 12 and 5) (Fig. 2b, Supplementary Data File 7). We also observed downregulation of metabolism associated processes, including fatty acids, sphingolipid de novo biosynthesis (cluster 8), cytochrome P450 (cluster 3) and ion transport by P-type ATPases (Cluster 9) during disease progression (Fig. 2a, b). GSEA and KEGG pathway analysis revealed correlated switches in expression patterns of genes involved in drug metabolism, fatty acid degradation, glycolysis and gluconeogenesis and glutathione metabolism (Supplementary Fig. 7a–d). To further explore changes during the progression to a progenitor like state we performed GSEA and GO ontology analysis when comparing progenitor like samples (Quartile 1 and 2) to differentiated like samples (Quartiles 3 and 4) (Supplementary Fig. 8a, Supplementary Data File 8). This also highlighted profound changes in metabolic processes with suppression of lipid metabolic processes being particularly prominent

(Supplementary Fig. 8a, b). Taken together, these findings indicate a switch from fatty acid and lipid metabolism to a glycolytic like state is likely to occur coincident with the acquisition of a progenitor like state.

**The orchestrated suppression of late epidermal differentiation and induction of progenitor-like gene expression is mediated by master regulators of epidermal differentiation**

The progressive differentiation of keratinocytes, from stem cell to corneocyte is governed by a complex network of transcriptional regulators including transcription factors (TFs) and long noncoding RNAs (lncRNAs). The selective expression of these key transcriptional regulators within the epidermal striatum demarcates regions of differentiation and progenitor self-renewal. Genetically non-redundant TFs including GRHL3, ZNF750, KLF4 and PRDM1 drive terminal epidermal differentiation[16,19–21] whereas, epigenetic transcriptional regulators such as PRMT1, ACTL6A, DNMT1 and EZH2 are essential to repress epidermal differentiation and maintain the progenitor state[22–25]. LncRNAs *TINCR* and *ANCR* (*DANCR*) have also been shown to act as important *trans* regulators of epidermal differentiation with *TINCR* functioning as an important driver of epidermal differentiation and *ANCR* acting as a progenitor-maintenance factor[26,27].

To determine whether the progenitor-like state observed in SCC is associated with the dysregulated expression of key drivers of epidermal differentiation we profiled the expression of several factors including *GRHL3*, *ZNF750*, *TINCR*, *KLF4* and *PRDM1* (Supplementary Fig. 9a). Transcriptional promoters of epidermal differentiation are significantly downregulated during SCC progression, whereas, TFs such as *PRMT1*, *ACTL6A* and *SMARCA5* that maintain the progenitor state in self-renewing somatic tissues[22,23,28]

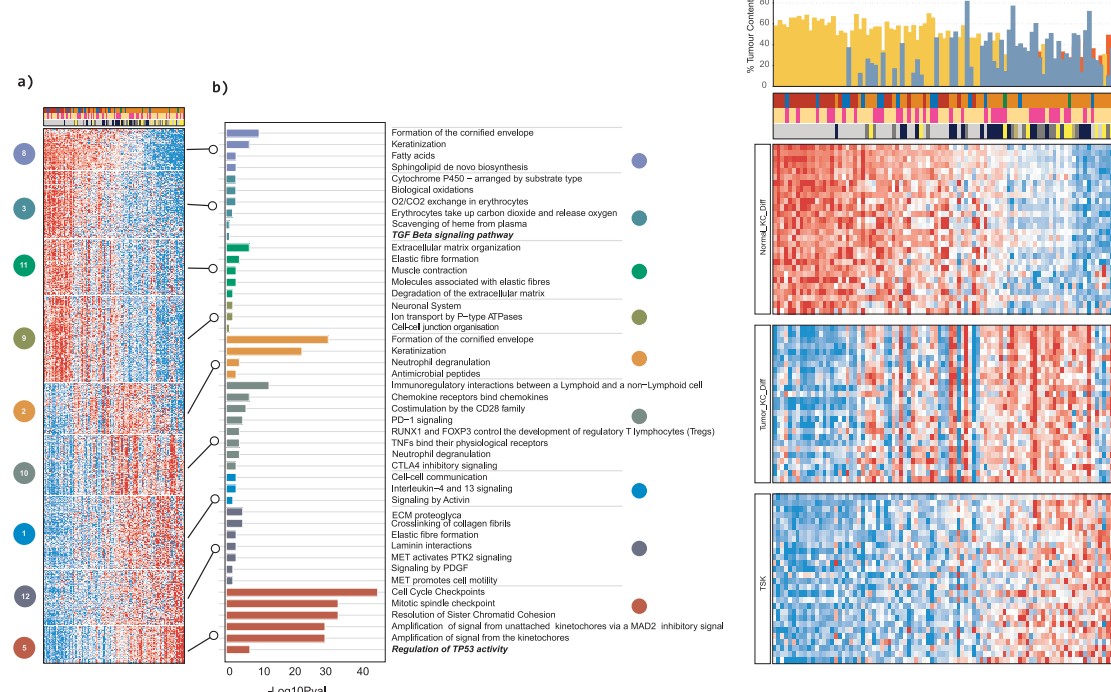

**Fig. 2 | cSCC progression is associated with the orchestrated gain and loss of key molecular pathways and/or processes associated with epidermal differentiation, cell-cell communication, metabolism, immune signalling and progenitor-like cell states. a** K-means clustering of normalised expression values identifies 15 core gene clusters representing co-ordinately expressed sets of genes. Heatmap shows gene expression levels of genes in 9 core co-expressed gene clusters with samples ordered by DvP signature score. **b** Gene set enrichment analysis showing significantly enriched molecular pathways and/or processes in the 9 core co-expressed gene clusters. Significance shown as bars -Log10(P values) Fischer's exact test (two-sided) adjusted for multiple testing. **c** Transcription factor regulon activities correlate with keratinocyte population dynamics. Sets of TFs regulons significantly (P values <= 0.05) correlated with percent enrichment of defined indicated single cell populations as estimated by CIBERSORTx. Samples are ordered by DP signature score with defined single cell enrichment estimates shown in the top bar chart (see Fig. 1g).

were significantly correlated with the progenitor-like state observed in SCC. The delineation of TFs according to differentiation status closely mirrored the selective expression of keratins (K), including K1, K10, K5 and K14 (Supplementary Fig. 9a). K1 and K10 genes are present in the late differentiation gene signature (Supplementary Data File 5) and their expression demarcates the intermediated spinous layer, whereas K5 and K14 are expressed in the basal layer. Gene expression scores of master regulators of differentiation (*ZNF750*, *KLF4*, *TINCR*) and associated signature scores of TINCR and STAU negatively correlated with DP signature score (Supplementary Fig. 9b, Supplementary Data File 9). In contrast, gene expression signature scores of *ACTL6A*, *DNMT1* and *PRMT1* positively correlated with DP signature score implicating them in controlling acquisition of the progenitor like state (Supplementary Fig. 9b).

To further define the set of TFs involved in SCC progression we performed TF network analysis. This analysis generates TF-centric regulatory networks by inferring the set of co-expressed genes associated with a given TF, thereby providing a link between the TF and a set of putative transcriptional targets collectively referred to as a "regulon". To identify regulons significantly enriched in either the Differentiated or Progenitor-like state, we performed VIPER (Virtual Inference of Protein activity by Enriched Regulon) analysis (Supplementary Data File 9). This identified 53 regulons (Pval <= 0.05) exhibiting significant Differentiated or Progenitor-like enrichment (Supplementary Fig. 9c, d). Coupled with TF Network analysis (Supplementary Fig. 9e) this highlighted TFs representing putative "master regulators" (MRs) of these two distinct transcriptional states. We next interrogated TF regulon expression dynamics in relation to keratinocyte population dynamics. This revealed significant positive correlations of TF regulons associated with NKD, TKD and TSK populations that aligned with the differentiated and progenitor like states (Fig. 2c) with activation of the TKD regulons pre-ceding activation of TSK TF regulons and downregulation of the NKD TF regulons.

**The orchestrated suppression of late epidermal differentiation and induction of a progenitor-like state is associated with dynamic changes in immune cell phenotypes**

K-means clustering analysis demonstrated that immune pathways are significantly enriched during SCC progression: in the progenitor like state neutrophil degranulation, immune checkpoint signalling via PD1 and CTLA4 and TNF signalling are significantly overrepresented (Fig. 2a). To further characterise the cSCC tumour immune microenvironment (TME) we performed cell type enrichment analysis using xCell[29]. This analysis identified 18 immune cell types significantly altered during disease progression and coincident with the acquisition of progenitor like characteristics (Fig. 3a; Supplementary Data File 10). This encompassed the initiation of enrichment of innate immune cell populations including dendritic cells (DC), neutrophils, monocytes, macrophages and plasmacytoid dendritic cells (pDC). Modulation of adaptive immune cells was also observed with enrichment of B cells and Treg populations followed with the additional loss of basophils, CD4+ Tem, CD4+ naïve T cells and CD8 + T cells and enrichment of Th2 cells (Fig. 3a).

Significant changes in expression patterns of immune inhibitory and stimulatory pathways were also observed coincident with immune cell population changes (Fig. 3a, Supplementary Fig. 10a; Supplementary Data File 10). Perhaps, most striking was upregulation of the immune "inhibitory" factors including *CD274*, (Fig. 3a) *BTLA*, *PDCD1*, *SLAMF7*, *IL10*, *TIGIT*, *CTLA4*, *LAG3*, *PDCD1LG2*, *IDO1* and *HAVCR2* (Supplementary Fig. 10a). This was followed by downregulation of ARG1 (Fig. 3a) and *EDNRB* and further enrichment of expression of *TGFβ1* (Supplementary Fig. 10a) and CD276 (Fig. 3a). Coincident with these changes was upregulation of immune "stimulatory" molecules including *CD28*, *CD27*, *CD40*, *CD80*; the chemokines *CXCL10*, *CXCL9*, and TNF pathway components including *TNF*, *TNFRSF1B, 4, 9, 14* and

*TNFSF4, 9*, as well as interleukins *IL1A* and *IL1B* (Supplementary Fig. 10a). These changes were reflected in immune pathway alterations including enrichment of scores for pathways and processes related to immune checkpoints (for example PD1_Data, CTLA4_Data), interferon and STAT1 signalling and interleukin signalling (Supplementary Fig. 10a). High correlations were observed across immune inhibitory genes (Fig. 3b) with enhanced expression of several of these correlating with enrichment of TKD and TSK populations (Fig. 3c). These data may explain the encouraging but limited response rates of cSCC to single agent cepilimumab anti-PD1 treatment[30,31] and suggests that combination checkpoint inhibition may enhance therapeutic responses.

Our sample set included 22 tumour samples from immunosuppressed patients enabling us to interrogate potential changes in the immune landscape between immunocompetent and immunocompromised patients. Several immune stimulatory and inhibitory pathways were significantly enriched in immunocompetent patients (Supplementary Fig. 10b, Supplementary Data File 10). These analyses also indicated that within both populations of immunosuppressed and immunocompetent patient subsets of samples could be described as immune active or immune silent showing high- or low-level expression of these genes respectively (Supplementary Fig. 10b).

**Modulation of The Epidermal Differentiation Complex precedes and accompanies dynamic changes in the immune landscape**

Interrogation of our k-means clustering and GSEA analysis revealed that increased expression of genes in cluster 2 precedes or is coincident with the dynamic changes in immunomodulatory pathways and processes (Clusters 1, 12). Cluster 2 is enriched for pathways involved in formation of the cornified envelope, keratinization, neutrophil degranulation as well as antimicrobial peptides (Fig. 2a). Many of the genes critical for cornified envelope formation and differentiation of the stratified epidermis are encoded by the epidermal differentiation complex (EDC) a ~ 2 Mb region on chromosome 1[32]. Horizon plot analysis indicated profound changes in EDC gene expression during cSCC progression (Supplementary Fig. 11; Supplementary Data File 10) and heatmap analysis revealed co-ordinated regulation of EDC genes with the DvP axis (Fig. 3d). Genes located in the 5′ region of the EDC include most of those encoding late cornified envelope (LCE) proteins and are highly expressed in differentiated samples and downregulated in the progenitor-like samples (Fig. 3d). Genes located more 3′ in the EDC include those which encode for the small proline rich proteins (SPRRs) and the S100 family of calcium binding proteins and are expressed at a low level in the most differentiated samples and are upregulated coincident with the emergence of the TKD cell population (Fig. 3d). Notably the most 3′ located EDC genes *S100A2* and *S100A6* are highly expressed in progenitor-like samples (Fig. 3d). This switch in 5′ to 3′ EDC gene expression patterns precedes and is coincident with the onset of changes in the immune cell complement and dynamic changes in immune modulatory gene expression (Fig. 3a, Supplementary Fig. 10a). These findings suggest that dynamic changes in EDC gene expression are associated with modulation of the immune landscape. Correlation analyses supported this hypothesis as we observed significant positive correlations between many of the 3′ EDC genes and immune cell populations (for example neutrophils, monocytes, pDC and Th2 cells) and immunomodulatory genes and processes (for example Type 1 interferon response, PD1_PDL1_Score, CD274, TGFβ1) enriched in progenitor like samples (Fig. 3e, Supplementary Data File 10).

**Driver gene combinations dictate disease progression**

The orchestrated sequence of events as disease progresses from a differentiated to a progenitor like state are likely to be dictated by the genetic changes caused by chronic UV damage. We have previously interrogated the genetic landscape of 12 of our primary tumour cSCC samples profiled by whole exome sequencing (WES)[11] (Fig. 4a). We

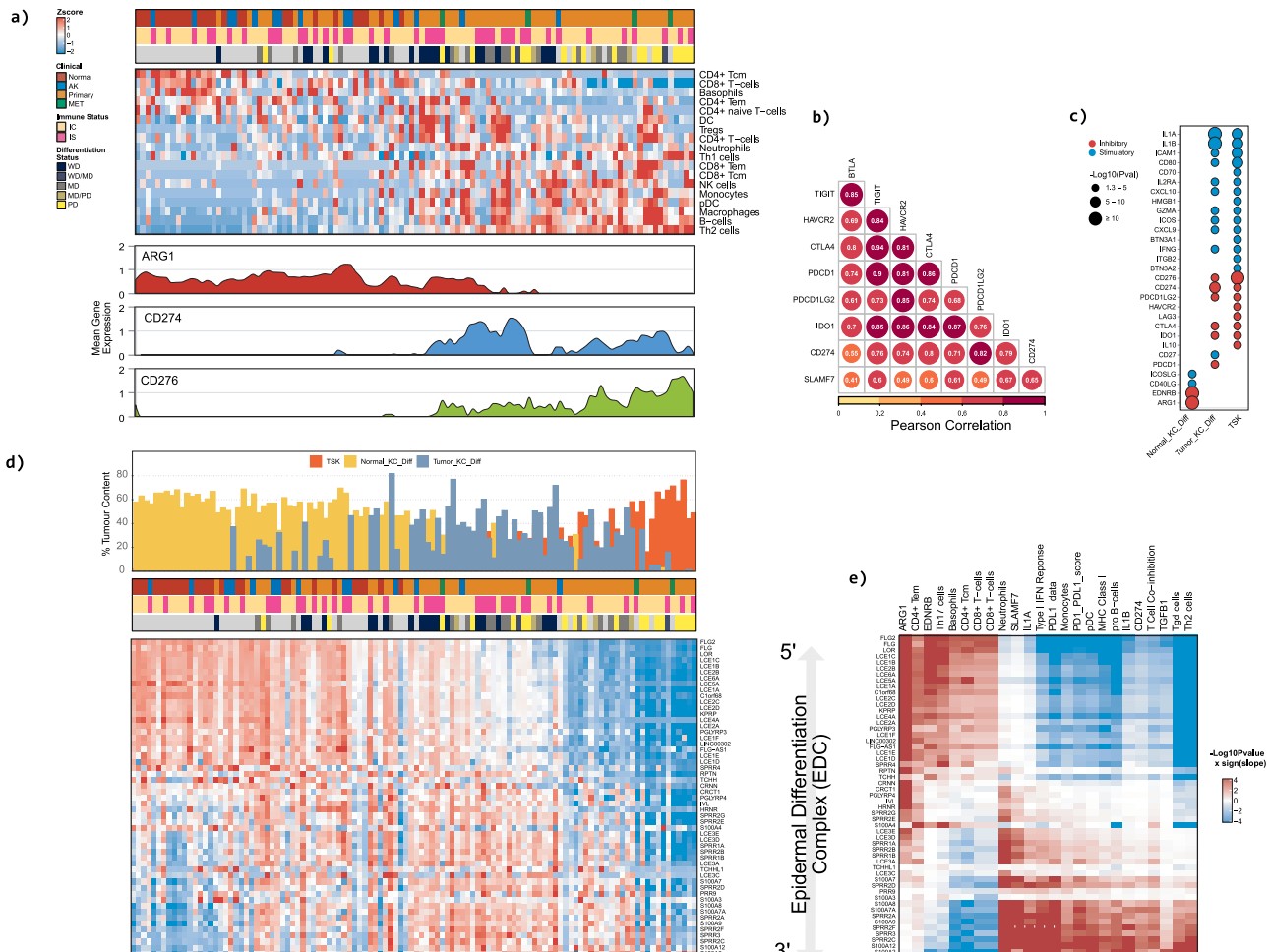

**Fig. 3 | The orchestrated suppression of late epidermal differentiation and induction of a progenitor-like state is associated with dynamic changes in immune cell infiltrates, immunomodulatory genes and correlates with epidermal differentiation complex modulation. a** Heatmap showing the relative enrichment of immune cell types and/or phenotypes as defined by xCell across the entire cSCC cohort. Area charts showing mean gene expression (lower panels) for cohort samples ordered by DP signature score. Mean *ARG1* expression is significantly downregulated as sample DP scores shift form late differentiation to progenitor-like. In contrast, the mean gene expression of immune inhibitory factors *CD274* and *CD276* show significant enrichment in samples with a high progenitor-like score. **b** Pearson correlation analysis of immune inhibitory factors. **c** Dot chart showing inhibitory and stimulatory immunomodulatory factors significantly correlated with Normal_KC_Diff, Tumour_KC_Diff and TSK enrichment bulk tumour

fractions. The size of each dot represents -log10(Cor *P*val) of the designated correlation. Significance was determined by two-sided Pearson's correlation test. *P* values were not adjusted for multiple testing. **d** Heatmap showing the relative expression of EDC genes. Samples are ordered by DP signature score and genes are ordered by DP signature correlation. Percent single cell enrichment estimates are shown in the top bar chart (see Figs. 1g, 2c). **e** Heatmap showing significant correlations between EDC genes and immune cell type/phenotype enrichment scores. Correlations are presented as -log10 (Cor *P*value) x sign (Cor) with red representing a significant positive correlation and blue representing a significant negative correlation. Pearson's correlations are shown in the plot. Significance was determined by two-sided Pearson's correlation test. *P* values were not adjusted for multiple testing. All correlations shown are significant.

found no correlation between DvP axis position and mutational burden (Supplementary Fig. 12a; Supplementary Data File 11). We next utilised CaSpER to identify and visualize copy number variation events by integrative analysis[33] of our bulk RNA-seq data of our AK and cSCC samples (Supplementary Fig. 12b, Supplementary Data File 11). This revealed a marked increase in copy number alterations observed in more progenitor like samples with a significant increase in the percentage of the genome altered with decreasing DvP quartile (Supplementary Fig. 12b, c, Kruskal-Wallis, $p = 4.5 \times 10^{-8}$). This is similar to our previous CNV analysis of WES data which indicated MD/PD tumours have a greater proportion of the genome altered by copy number variations compared to WD tumours[11]. Notable copy number losses of 3p and 9p and gains of 19p were observed in both analyses (Supplementary Data File 11). 26% of genes affected by copy number loss (83 out of 315) also showed a significant down regulation of gene expression (*p*adj<0.05) when comparing Quartile 1 and Quartile 2 samples to

Quartile 3 and Quartile 4 samples indicating a potential tumour suppressor role for these genes (Supplementary Data File 11). Many immunoglobulin heavy chain genes showed copy number gains and an elevation of gene expression likely reflecting enhanced B cell infiltration in progenitor like samples. Excluding these, 31% of genes (14 out of 45) showed both a copy number gain and a significant elevation of gene expression in progenitor like samples indicating potential tumour promoter roles (Supplementary Data File 11).

Oncoprint analysis of functionally validated/implicated genetic drivers of cSCC development of our samples previously also profiled by WES revealed the possibility that combinations of driver gene events may determine disease progression (Fig. 4a; Supplementary Data File 11). We observed early loss of the tumour suppressor genes *TP53*, *NOTCH1*, *TGFBR1*, *TGFBR2* and subsequent activation of MAPK signalling through mutational activation of KRAS (KRAS$^{G12D}$ mutation in PD02) or HRAS (HRas$^{Q61L}$ mutation in PD07). Analysis of 151 cSCC

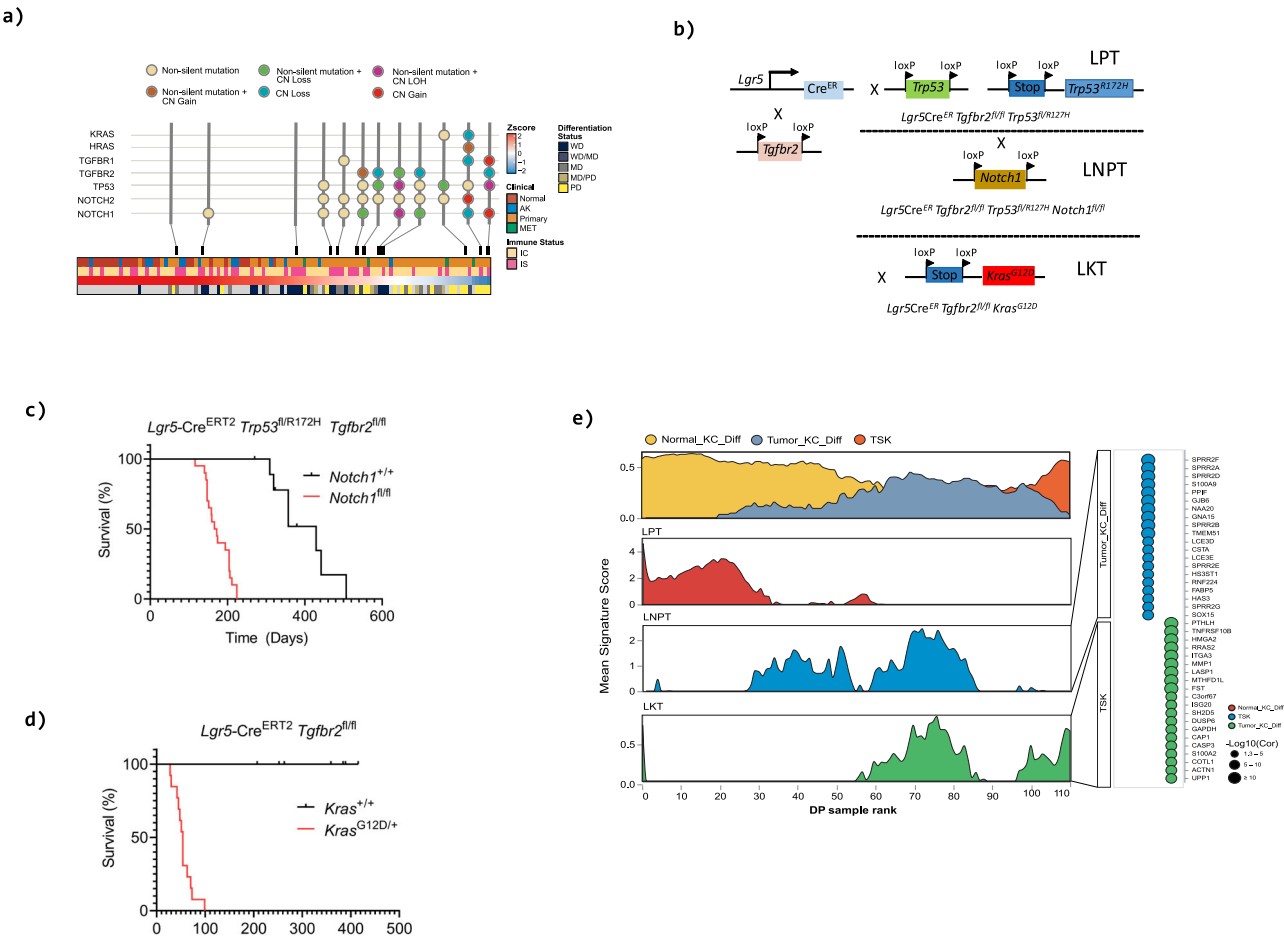

**Fig. 4 | Driver gene combinations dictate disease progression in murine genetically engineered cSSC and recapitulate human disease progression.**
**a** Oncoprint of selected driver genes in samples profiled by whole exome sequencing[11] ordered by DP axis rank. **b** Schematic description of genetic crossing strategies. Cre, cre recombinase, ER, estrogen receptor, loxp, Cre-lox recombination site. **c, d** Kaplan-Meier analysis of overall survival. Loss of *Tgfbr2* coupled with mutation/loss of *Trp53* drives skin tumorigenesis in mice (LPT, *n* = 9; L = *Lgr5*, P = *Trp53*, T = *Tgfbr2*) which is accelerated by loss of *Notch1* (LNPT, *n* = 20, N=*Notch1*), (*p* < 0.001 [log rank Mantel-Cox test, chi square 25.13, df 1]) (**c**). Combinatorial

knock in of activated Kras[G12D] coupled with deletion of *Tgfbr2* results in rapid skin tumour formation (LKT, *n* = 13, LT *n* = 16, K=Kras[G12D]) (**d**). **e** Area charts showing mean tumour cell type enrichment (top panel) and mean GEMM signature enrichment (lower panels) for cohort samples ordered by DP signature score. Genes significantly enriched (*P*val <= 0.05 and logFC >= 2) in a specific mouse genotype were used as signature genes for enrichment analysis. Single sample gene set enrichment (ssGSEA) was employed to determine signature enrichment in bulk human cSCC. Source data for **c** and **d** are provided in the Source Data file.

tumour samples curated in Cbioportal[34,35] indicated TP53 mutations in 76%, NOTCH1 in 55%, *TGFBR1* in 6%, *TGFBR2* in 8% and mutation of *KRAS*, *HRAS* and *NRAS* combined occurred in 17.6% of samples (Supplementary Fig. 13). Consistent with involvement of these signalling pathways in mediating disease progression we observed significant negative correlation of TGFβ, NOTCH1 and TP53 signalling signatures and positive correlation of ERK signalling signatures with DvP signature score with selective modulation of signalling components occurring with disease progression including downregulation of *TP53*, *TGFBR2* and *NOTCH1* in progenitor like samples (Supplementary Fig. 14; Supplementary Data File 12).

We and others have previously demonstrated that murine cSCCs that phenotypically resemble human cSCCs can efficiently initiate from the hair follicle Lgr5+ve stem cell compartment[36,37] and that tumours originating from these cells can show aggressive features[38-40] potentially enabling us to model a broad spectrum of the cSCC disease continuum. Therefore, to functionally validate these genetic events as drivers of disease progression we targeted combinatorial inactivation of *Tgfbr2*, *Notch1*, *Trp53* and mutational activation of Kras and Trp53 to murine Lgr5+ve stem cells using *Lgr5-EGFP-Ires-Cre[ERT2]*, *Tgfbr2[fl/fl]*, *Notch1[fl/fl]*, *Trp53[fl]*, LSL-*Trp53[R172H]* and *LSL-Kras[G12D]* mice (Fig. 4b; Supplementary Data

File 13). Loss of *Notch1* alone or in combination with loss of *Tgfbr2* or the combination of loss and mutation of *Trp53* did not result in significant skin tumour formation (Supplementary Data File 13). Loss of *Tgfbr2* coupled with Trp53loss/mutational activation (LPT mice) resulted in skin tumour formation with long latency (median survival = 429 days) (Fig. 4c). Additional loss of *Notch1* (LNPT mice) greatly accelerated skin tumour formation (median survival =169 days, *p* < 0.001 [log rank Mantel-Cox test, chi square 25.13, df 1]) (Fig. 4c). Mutational activation of *Kras* alone did not result in skin lesions, nor did deletion of *Tgfbr2*, whereas the combination of *Kras* mutation coupled with loss of both alleles of *Tgfbr2* resulted in rapid skin tumour formation (LKT mice, Fig. 4d, Median survival 54 days) consistent with our previous observations on deletion of *Tgfbr1*[36]. The differentiation status of selected tumours harvested at endpoint from these cohorts was assessed by pathological analysis of haematoxylin and eosin (H&E) stained sections from formalin fixed and paraffin embedded tumours (Supplementary Fig. 15, Supplementary Data File 13). Most tumours were moderately differentiated irrespective of genotype.

We performed bulk RNASeq analysis of snap frozen murine tumours harvested at endpoint from our genetically engineered mouse models (GEMMs) (Supplementary Data File 14). Genes

significantly enriched (*P*val <= 0.05 and logFC >= 2) in each specific mouse genotype were used as signature genes for further enrichment analysis (Supplementary Data File 14). We employed single sample gene set enrichment (ssGSEA) to determine murine signature enrichment in the human samples to disease position our GEMMs relative to the human DvP axis and tumour cell populations. This analysis revealed that LPT tumours represent early stages of the human cSCC continuum (Fig. 4e, Supplementary Data File 14). Several LNPT tumour signature genes show significant correlation with TKD genes, and these tumours correspond with human samples spanning the middle of the cSCC continuum (Fig. 4e). Several LKT tumour signature genes show significant overlap with several TSK genes and these tumours correspond to human samples which are more progenitor like (Fig. 4e).

### Conservation of dynamic changes in transcription factor expression, EDC modulation and immune infiltration in murine and human tumours

GSEA analysis revealed pathways and processes significantly altered between our murine tumour genotypes (Supplementary Fig. 16a; Supplementary Data File 14). Comparisons of LNPT with LPT mice revealed alterations of GO terms associated with epidermal development, neutrophil migration and cell signalling pathways and corresponding changes in gene expression of TFs, cornification, immunomodulatory and signalling pathway genes associated with these processes (Supplementary Fig. 16b). Comparison of LKT tumours with LNPT and LPT tumours revealed further changes in cell differentiation, metabolism and immune processes with associated changes in gene expression of many factors regulating these events (Supplementary Fig. 16a–c). Differential gene expression analysis between our GEMMs revealed dynamic changes in transcription factor expression with notable changes in expression of NKD enriched factors *Dlx5* and *Tfap2c*, TKD enriched factors *Hif1a* and *Mafb* and TSK enriched factors *Klf7* and *Fosl1* that also exhibited dynamic modulation along the DP axis (Fig. 5a, Supplementary Fig. 16c).

These analyses indicated that the changes we observe in human samples that accompany disease progression are recapitulated in our murine tumours. The murine EDC is encoded by a ~ 3.5 Mb region of Chromosome 3 with many genes syntenic with their human counterparts[41] (Supplementary Fig. 15d). We observed significant changes in subsets of EDC genes between genotypes consistent with dynamic changes in cornification/keratinisation genes (Supplementary Fig. 16c, d; Supplementary Data File 14) reminiscent of changes we observe in human cSCC. We also observed many significant changes in immune and inflammatory response genes across murine tumour genotypes (Supplementary Fig. 16c) with notable correlations of EDC genes and immune cell markers (Fig. 5b) such as *S100a2* that we also observed in human samples (Fig. 5c, d and Fig. 3e). We performed immunohistochemical analysis of immune cell markers in our murine tumours to evaluate changes in immune cell population tumour infiltration associated with disease progression and driver gene combinations (Supplementary Fig. 17). All tumours irrespective of genotype displayed higher numbers of immune cells surrounding the periphery of tumours (border) when compared to numbers infiltrating the tumours (Fig. 5e). With disease progression from LPT to LNPT to LKT genotypes we saw a decrease in CD3+ve T lymphocytes and significant decreases in the number of CD4+ve T cells present both within and at the margins of tumours (Fig. 5e, Supplementary Date File 15). A similar trend in CD8+ve T cells was also observed (Fig. 5e). Macrophage populations (F4/80 +ve cells) were variable but tended to increase in LKT tumours and neutrophil populations (Ly6g +ve) decreased between LNPT and LPT tumours (Fig. 5e). Comparison of the ratio of adaptive (CD4+ve and CD8+ve T cells) to innate immune cells (F4/80 and Ly6G +ve cells) indicated a potential switch from an adaptive to an innate like immune complement coincident with disease progression (Fig. 5f). Remarkably, a similar assessment of xCell data from the

human RNAseq samples with whole exome data revealed the same switch with analogous genotypes (Fig. 5f, Supplementary Data File 11).

Murine tumours induced by solar UV radiation of hairless mice recapitulate the histopathical features of human cSCC disease progression and are similarly genetically complex[42]. Intersection and Uniform manifold Approximation and Projection (UMAP) of RNaseq data from a UV radiation treated hairless mouse model[8] with our genetically induced murine tumours indicates a remarkable overlap of disease trajectories (Supplementary Fig. 18, Supplementary Data File 16). Our LPT tumours represent very early disease and overlap with chronic UV damaged skin (CHR), LNPT tumours span early disease from CHR to papilloma formation and LKT tumours overlap with invasive SCC in the UV model. This analysis provides further evidence that the underpinning driver gene events dictate disease progression regardless of the genetic complexity in which they take place and the causative mechanism which generates them.

Given that the genetic events in our GEM models are targeted to the murine Lgr5+ve hair follicle stem cell compartment and recapitulate the human disease continuum we sought to mine our human data for potential indicators of hair follicle stem cell fate. Deconvolution of our bulk cSCC RNAseq data using human hair follicle cell state transcriptional signatures representing 23 cell types/states[43] (Supplementary Fig. 19a, Supplementary Data File 17) revealed down regulation of interfollicular epidermis (IFE) granular and spinous.3 signatures with acquisition of a progenitor like state (Supplementary Fig. 19b, Supplementary Data File 17), consistent with our loss of differentiation signatures described above. Concomitant with these changes were notable increases in the IFE basal.2, IFE mitotic, lower bulge and outer root sheath suprabasal (ORS.SB) signatures (Supplementary Fig. 19b, Supplementary Data File 17). We observed significant increases in the proportion of epithelial cells with an IFE basal.2 signature with transition from normal skin to AK and primary tumour to metastatic disease states (Supplementary Fig. 19c) and progression from DvP quartile 3 to 2 (Supplementary Fig. 19d). Similarly, we observed significant increases in the percentage of epithelial cells with an IFE mitotic signature from normal skin to AK and from AK to primary tumour disease states (Supplementary Fig. 19e) and a stepwise increase in their prevalence during progression through the DvP quartiles (Supplementary Fig. 19f). We also observed a significant increase in the percentage of epithelial cells with a lower bulge signature in metastatic samples (Supplementary Fig. 19g) and with transition through DvP quartiles (Supplementary Fig. 18h). Primary tumour samples were also enriched for epithelial cells with an ORS.SB signature (Supplementary Fig. 19i) and these were also observed in the most progenitor-like samples (Supplementary Fig. 19j). Recent studies indicate that both the lower bulge and ORS.SB cells express Lgr5[44] (Supplementary Fig. 19a) implicating the possible involvement of Lgr5+ve cells as well as IFE cells in human cSCC initiation/progression. Taken together our findings suggest that regardless of the mechanism that generates driver gene alterations or the cells in which they take place it is the driver gene combinations that dictate cSCC disease progression.

## Discussion

cSCC is the second most common skin cancer worldwide[1] with incidence rates increasing by 5% per year in the UK[45]. The genomic landscape of cSCC is dominated by tumour suppressor gene loss and contains few potentially actionable oncogenic events[10]. The frequency and diversity of genetic alterations found in UV exposed skin, AK and cSCC has further hampered rational therapeutic strategy development. Similarly, risk-stratified deployment of currently available therapeutic and adjuvant strategies has been impacted by the limited prognostic utility of current clinicopathologic staging systems. With restricted treatment options for locally advanced or metastatic cSCC once surgery/radiotherapy has failed, there is also an urgent need for new targeted treatments or immunotherapeutic approaches. This is

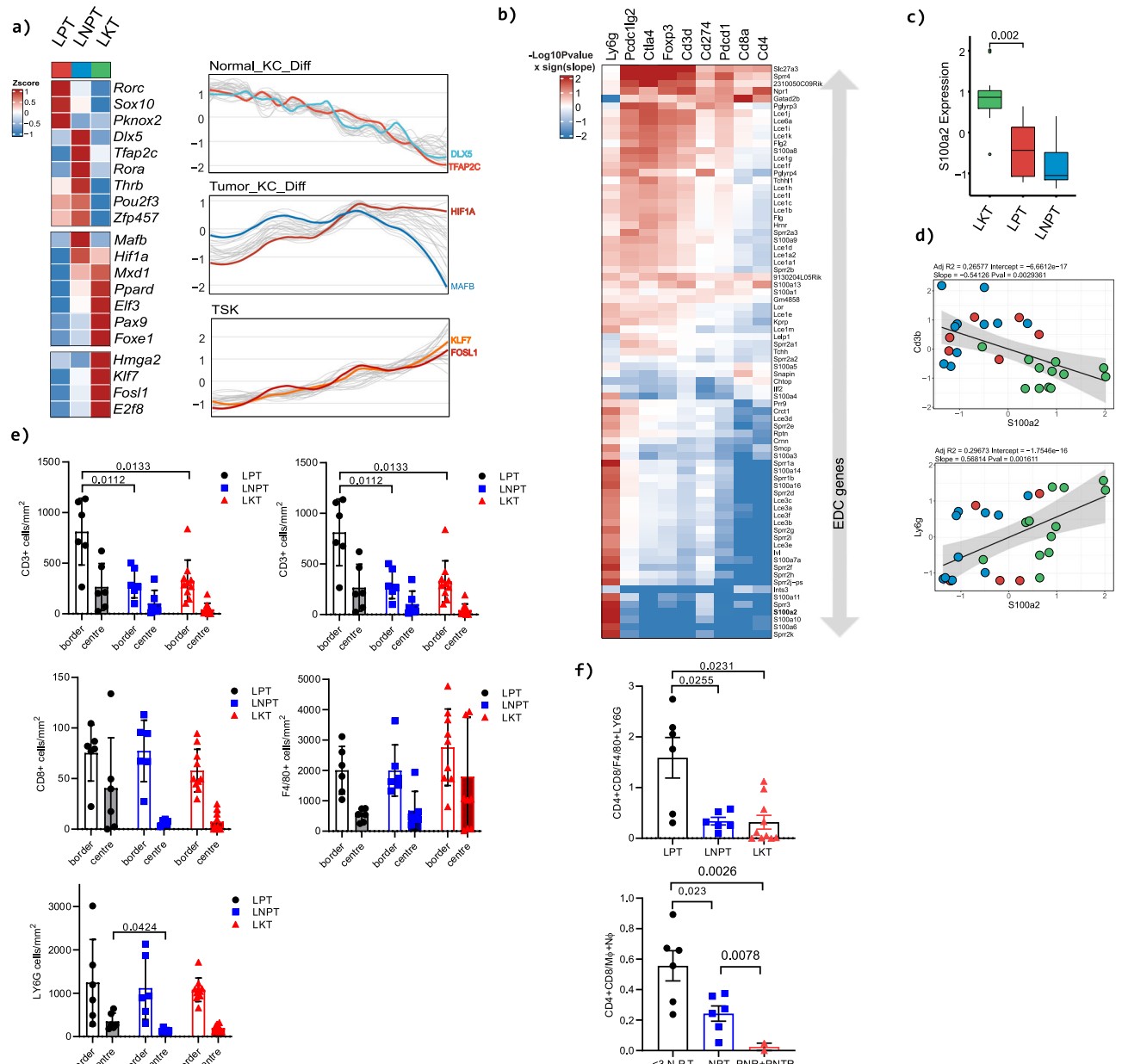

**Fig. 5 | Conservation of transcription factor regulation, EDC modulation and immune modulation in murine and human tumours. a** Heatmaps of transcription factors significantly enriched in the indicated genotypes (left panel). Transcription factor enrichment plots in indicated human keratinocyte samples ordered by DP axis. Selected conserved TFs between murine genotypes and human keratinocyte populations are highlighted. **b** Heatmap showing significant murine EDC gene correlations with immune cell marker gene expression. **c** Box plot showing *S100a2* expression in indicated murine tumours (LKT, *n* = 12; LPT, *n* = 6; LNPT, *n* = 12; L=*Lgr5*, P=*TrpS3*, T=*Tgfbr2*, N=*Notch1*, K= Kras[G12D]). Plot is annotated with Wilcoxon rank sum *P* value (two-sided) not adjusted for multiple testing. **d** Scatter plots of *Cd3b* and *Ly6g* immune cell marker gene expression versus *S100a2* in indicated murine tumours. Pearson's correlations are shown in the plots. Significance was determined by two-sided Pearson's correlation test. *P* values were not

adjusted for multiple testing. The plots show a solid regression line and error bands representing 95% confidence intervals. **e** Bar charts of immune cell populations measured by IHC in GEMM models of CD3, CD4, CD8, F4/80 and Ly6G positive populations. (LPT *n* = 6; LNPT, *n* = 6; LKT, *n* = 10; shaded bars indicate tumour centre, empty bars tumour border). Mean +/- SD are shown. *=*p* < 0.05, **=*p* < 0.01, 2-tailed Welch's *t* test. **f** Conservation of changes in the ratio of adaptive (CD4 and CD8 +ve T cells) to innate immune cells (macrophages and neutrophils) assessed by IHC in murine tumours (upper box plot) and human tumours assessed by Xcell (lower box plot) of similar genotypes, *TP53* mutation (P), *NOTCH* 1 or 2 mutation (N), *TGFBR1*/*TGFBR2* mutation (T), HRAS or KRAS mutational activation (R). Mean +/− SD are shown (<3 N, P, T *n* = 5; NPT, *n* = 5; PNR + PNTR, *n* = 2) *=*p* < 0.05, **=*p* < 0.01, 2-tailed Welch's *t* test. Source data for c, e and f are provided in the Source Data file.

particularly important for immunosuppressed organ transplant recipients for whom cSCC are twice as likely to prove fatal[3] and in whom checkpoint inhibitor immunotherapy is relatively contraindicated because of high risk of allograft rejection. These many unmet clinical needs and challenges in cSCC management require a much greater understanding of its biological basis and the driving events of disease progression.

Here we employed a systems biology approach to integrate bulk RNASeq analysis of the largest cohort of human samples to date with prior WES and single cell sequencing analysis coupled with integration of genetically engineered murine models. We found that whilst lesions broadly cluster with clinicopathologic definitions many samples cluster atypically, notably including several AK samples which cluster with the majority of cSCC and metastasis samples. These may represent AKs with

a high chance of progression to cSCC. We therefore conclude that cSCC disease progression is perhaps best classified as a disease continuum in which lesions fall on a spectrum from more differentiated to a progenitor-like state. Accompanying this progression are changes in the frequency of recently identified distinct keratinocyte populations[17] with the epithelial compartment of the most progenitor-like samples comprising 60–80% TSKs. Deconvolution analyses also revealed increases in epithelial cells with gene expression signatures of IFE and Lgr5+ve derived cell populations during disease progression indicating potential tumour initiation in these cell compartments.

Mutations in driver genes, including *NOTCH* family members and *TP53* are detected at high frequency in apparently normal sun-exposed skin. In fact, it is estimated that putative driver mutations exist at a density of ~140 driver mutations per square centimetre[12]. However, while these mutations may drive skin cells one step closer to malignancy, findings presented herein demonstrate that cSCC progression necessarily involves defined combinations of driver mutations illustrated here with those targeting NOTCH, TGFβ, TP53 or RAS-MAPK pathways. These findings suggest a model in which the acquisition of combinatorial NOTCH, TGFβ, TP53, RAS-MAPK mutational states confers phenotypic advantages that promote the clonal expansion of basal-like (progenitor) cells leading to patches of cells (lesions) that are phenotypically altered. Recent studies comparing mutational rates of *NOTCH1* in normal human skin and cSCC samples have suggested that NOTCH1 may not contribute to transformation[13] and in the esophagus NOTCH1 loss promotes clonal expansion but can impair tumour growth[46]. Our studies here clearly demonstrate that in the murine skin at least, NOTCH1 has tumour suppressor function. We did not observe any correlation with mutational burden and human disease progression but did observe increased CNA alterations in more progenitor like samples compared to more differentiated samples. This may reflect loss of TP53 leading to genome instability as recently observed in pancreatic ductal carcinoma[47], but, if this is the case and if genome instability per se contributes to disease progression in cSCC warrants further investigation. Our findings also reconfirm a tumour suppressor role of TGFβ signalling in cSCC[36]. Consistent with this, recent studies in organotypic cultures indicate that loss of TGFβ signalling may promote keratinocyte invasion[48]. It is important to note that whilst we observe correlation with loss of some TGFβ signalling pathway components with disease progression it is unlikely that this is obligate for cSCC formation and TGFβ signalling may also play tumour promoting roles by promoting EMT and/or immune escape[49] and this requires further investigation in cSCC. Importantly, our findings suggest that immune changes driven by combinations of somatic mutations promote the selection and expansion of mutant clones with keratinocyte differentiated cell states associated with significant inflammation and progenitor-like (basal) states associated with immune escape.

In addition, we demonstrate that changes in transcription factor networks are dynamically modulated during human and murine disease progression and may control tumour growth and progression towards a progenitor like state. Indeed, recent evidence has implicated FOSL1 as important for proliferation of human SCC cell lines[17]. Transcription factors such as MAFB and FOSL1 are also likely master regulators of EDC genes which exhibit dynamic changes in expression both early and throughout disease progression. How dysregulated NOTCH and TGFβ signalling cascades intersect with these transcriptional programmes requires further investigation.

We observed profound modulation of the immune landscape during progression along the DvP axis in both human and murine samples with concomitant expression of immunomodulatory genes and a switch from an adaptive to innate like immune profile with the acquisition of a progenitor like state. The changes we observe in immune cell populations in our human samples identified by xCELL and in murine tumours by immunohistochemistry suggest a possible tumour promoting role in cSCC disease progression, but this requires

future further in-depth analysis and functional interrogation. The EDC genes themselves may directly influence the immune landscape of tumours. S100 genes are known modulators of inflammation[50] and our data implicating members of this family in cSCC disease progression requires further investigation. Recent clinical employment of PD-L1 immune checkpoint blockade in locally advanced and metastatic cSCC has shown promise in a subset of cSCC[30,31]. Here we observe frequent co-expression of multiple checkpoint molecules indicating that combination immune checkpoint therapies may be more effective in management of disease. Our cross-species analysis indicates that driver gene combinations directly influence disease progression and, in agreement with previous studies[8,51], that progressive activation of ERK signalling provides a potential therapeutic target for cSCC. Finally, further combinatorial analyses are required to indicate if panel-based DNA analysis coupled with gene expression profiling may have utility in predicting disease outcome in patients.

In summary, our study provides a framework and resource which can be interrogated and exploited to not only understand the pathomechanisms of disease progression and improve prognostication but also to further identify potential therapeutically actionable vulnerabilities.

## Methods

### Collection of patient samples

This study was approved by the East of Scotland Research Ethics Service (REC reference 08/S1401/69), The Ethics and Scientific Committee of A. Sygros Hospital (Ref 2353/3-11-2016) and The University of California, San Francisco Institutional Review Board and was conducted according to the Declaration of Helsinki Principles. All patients participating in the study provided written, informed consent. Punch biopsies of samples were snap frozen in liquid nitrogen.

### Human total RNA isolation

Frozen tissue was homogenised on dry ice using a 2 ml Kimble dounce tissue grinder set (Sigma-Aldrich, D9063), suspended in RNA lysis buffer from the Qiagen RNeasy Micro Kit (Qiagen, 74004) and passed 10 times through an 18 gauge syringe needle before proceeding with RNA isolation according to the manufacturer's protocol including the optional DNA degradation step using the Qiagen RNase-Free DNase kit (Qiagen, 79254). RNA quality was assessed on the Agilent 2100 bioanalyser using the RNA 6000 nano kit (Agilent, 5067-1511).

### Mouse total RNA isolation

Skin tumours were harvested and bisected with half placed into RNA*later* and snap frozen on dry ice. Tissue was homogenized using the Precellys lysing kit (Bertin Instruments, KT03961-1-003-2) in a Precellys Evolution tissue homogenizer. RNA was isolated using the Qiagen RNeasy Mini Kit (Qiagen, 74104) according to the manufacturer's protocol, including the optional DNA degradation step using the Qiagen RNase-Free DNase kit (Qiagen, 79254). RNA integrity was analysed with a NanoChip (Agilent RNA 6000 Nanokit; 5067-1511).

### RNA sequencing

Human RNA sequencing analysis was performed as previously described[52]. Briefly, the TruSeq Stranded Total RNA protocol (part no. 15031048 Rev. D April 2013) was used to generate sequencing libraries from 500ng-1ug total RNA. RNAseq libraries were sequenced on the HISeq2000 platform. Murine RNA sequencing libraries were prepared from 2 µg of poly(A) selected RNA and were sequenced on an Illumina NextSeq 500 sequencing system using the High-Output kit (75 cycles). RNA sequencing raw read data was analysed using the nf-core/rnaseq pipeline[53,54]. Sequencing reads were mapped to human genome assembly GRCh38 or mouse genome assembly GRCm38 using STAR aligner[55] to generate quality control (QC) metrics and gene counts. Samples passing gold standard QC metrics were retained for further

analysis. Gene counts representing individual tumours were subsequently processed using the DESeq2 R package[56,57]. Normalised LogR values were generated from gene count data for downstream clustering and statistical analyses. Differential Gene Expression (DGE) analysis was performed using standard DESeq2 workflows.

## Statistical plotting

Boxplots and bar plots were generated by R packages ggpubr[58] and bbplot[59]. Correlation plots were generated using the ggplot2 R package[60,61]. Heatmaps were generated using ComplexHeatmap[62] and circlize[63] packages in R. Python graphing library Altair[64] was used to generate gene and GEMM signature density plots as well as immunoregulatory bubble charts. Horizon plots showing EDC gene expression were generated using R package Gviz2[65]. Correlation heatmaps were generated using the corrplot R package[66].

## Clustering analysis

LogR normalised gene expression values were used for patient clustering. Hierarchical, PCA and tSNE analyses were implemented by the R packages, FactoMineR[67], factoextra[68] and Rtsne[69], respectively. The 2000 most variable genes as determined by median absolute deviation of the LogR normalised counts data were used for unsupervised clustering.

## Gene set enrichment analysis (GSEA) – human and mouse RNAseq data

GSEA was performed using genes significantly and differentially expressed between the indicated clinical designations or genetic backgrounds. A cutoff of padj <= 0.05 & abs(log2FoldChange) >= log2(2) was used to select genes for GSEA. The clusterProfiler[70], ReactomePA[71], msigdbr[72] and/or dnet[73,74] R packages were used for GSEA. Single sample gene set enrichment analysis (ssGSEA) was performed using the GSVA[75,76] and genefu[77] R packages. ssGSEA scores calculated for each sample were used for downstream statistical analysis and visualisation.

## k-means clustering

Hierarchical k-means clustering was performed using normalised gene expression representing the 2000 most differentially expressed (unique) genes between clinical designations i.e normal versus primary, normal versus AK, AK versus primary using padj <= 0.05 & abs(log2FoldChange) >= log2(2) as the gene selection cutoff. Hierarchical k-means clustering was implemented by the hkmeans function in the factoextra R package[68]. The number of k clusters was estimated by "gap" statistic using the cluster R package[78]. The enrichment of a given k-means cluster in a patient sample was determined using ssGSEA as implemented by the GSVA R package[75]. Individual k-means signature scores were then correlated with the DvP score. k-means clusters were ranked according to their correlated enrichment scores and visualisation.

## Generation of differentiation versus progenitor (DvP) score

The DvP score was generated using early, late differentiated gene signatures and progenitor gene signature sets derived from organotypic models of epidermal differentiation[16]. Normalised gene expression values, representing the combined set of signature genes, were clustered using the R package ConsensusClusterPlus[79]. This analysis identified two stable clusters of patient samples which were subsequently used as classes to define a signature of significantly and differentially expressed genes representing both the differentiated and progenitor-like states. This set of genes is referred to as the DP signature gene set. The DP signature gene set was used to generate a signed score (i.e., DP Score) using the function sig.score from the R package genefu[77] and to rank patients along a continuum of differentiation.

## RTN analysis

Transcription Factor network inference was performed using the RTN R package[80] as described in[81]. Master regulators were identified using the *msviper* function as implemented by the R viper package[82]. TF regulon activity was calculated for each sample using the *viper* function. Inferred TF regulon activity scores were used for downstream statistical analyses and visualisation. Regulon networks were visualised using the R package RedeR[83].

## CIBERSORTx and deconvolution analysis

CIBERSORTx analysis was implemented using the web framework located at https://cibersortx.stanford.edu, which provides detailed instructions for data input and computation[18]. Briefly, single cell RNA-seq data representing individual normal and cSCC cell populations obtained from Ji et al.[17] were used to define a signature matrix consisting of barcode genes that discriminate normal and neoplastic cell subsets of interest. This signature matrix was then applied to bulk RNA expression profiles to infer cell type proportions within each patient sample. Inferred cell type proportions were used for downstream analysis. Deconvolution of bulk cSCC RNAseq data using human hair follicle cell state transcriptional signatures[43] was performed using the Gene expression deconvolution interactive online tool (GEDIT) found at: http://webtools.mcdb.ucla.edu (Default settings). Cell state enrichment scores for each patient sample were plotted in a barplot either together (relative enrichment) or individually. Deconvolution of normalised gene expression was also performed by the xCELL package in R[29] to identify immune cell types and/or phenotypes enriched in patient samples. Immune scores generated by the xCELL algorithm were used for downstream statistical analysis and plotting.

## Copy number variant analysis

Copy number variation analysis was carried out on bulk human RNAseq data using CaSpER[33] (with default settings). RNAseq data from normal sun exposed skin or normal perilesional skin were used as controls. Final copy number data was plotted in R using ggplot2.

## GEMM signature and cSCC enrichment analysis

LKT, LPT and LNPT gene signatures were generated by filtering representative differentially expressed genes exhibiting an adjusted *P* value <= 1e-3 and Log2 Fold Change >=1. Selected mouse signature genes were then converted to a supported human gene ortholog using the msigdbr R package[72]. Orthologous LKT, LPT and LNPT human gene signatures were used to perform ssGSEA as described above to generate patient specific enrichment scores. ssGSEA enrichment scores were subsequently used for downstream statistical analysis and visualisation.

## Murine clustering analysis

UV induced mouse cSCC data[8] was intersected with scaled GEMM RNAseq data in R, Uniform Manifold Approximation and Projection (UMAP) was carried out on the combined data within R using the package UMAP labelled with directlabels and visualised using ggplot2[60,84].

## Murine studies

All animal experiments were performed in accordance with UK Home Office regulations (project licence 70/8646), and adherence to the ARRIVE guidelines, and were reviewed and approved by the Animal Welfare and Ethical Review Board of the University of Glasgow. Mice used were segregating for C57BL/6J and S129 background. Alleles used throughout this study were: *Lgr5-cre-ER*[T2][85], *Tgfbr2*[fl/fl][86], *Kras*[G12D][87], *Tp53*[fl], *Tp53*[R172H][88], *Notch1*[fl/fl][89]. A mix of males and females were used. Recombination was induced with one intraperitoneal injection of 3 mg Tamoxifen (Sigma) followed by one injection of 2 mg Tamoxifen daily for three days. Mice were induced at 6–15 weeks of age, monitored thrice weekly and humanely culled at clinical endpoint defined by

tumour burden. Mice were censored at 550 days after initial Tamoxifen injection. Mice of both sexes were included in the ageing cohorts.

## Immunohistochemistry

Murine tumour tissues were fixed in 10% neutral-buffered formalin overnight and embedded in paraffin. All Immunohistochemistry (IHC) staining was performed on 4 μm formalin fixed paraffin embedded sections (FFPE) which had previously been heated at 60 °C for 2 h. The following antibodies were used on a Leica Bond Rx autostainer: CD3 (Abcam, ab16669, clone SP7, 1:100), CD4 (eBioscience, 14-9766-82, clone 4SM95 1:500), CD8 (eBioscience, 14-0808-82, clone 4SM15, 1:500), F4/80 (Abcam, ab6640, clone A3-1, 1:100) and LY6G (BioXcell, BE0075-1, clone RB6-8C5, 1:60,000). All FFPE sections underwent on-board dewaxing (Leica, AR9222) and antigen retrieval using appropriate retrieval solution. Sections for F4/80 staining underwent antigen retrieval using enzyme 1 solution (Leica, AR9551) for 10 minutes at 37 °C. Sections for CD3, CD4, CD8, and Ly6G underwent antigen retrieval using ER2 solution (Leica, AR9640) for 20 min at 95 °C. Sections were rinsed with Leica wash buffer (Leica, AR9590) before peroxidase block was performed using an Intense R kit (Leica, DS9263). Sections for CD4, CD8 F4/80 and LY6G had the blocking solution applied from the Rat ImmPRESS kit (Vector Labs, MP-7404) for 20 min. Sections were rinsed with wash buffer and then the primary antibody applied at the optimal dilution (CD3, 1/100; CD4, 1/500; CD8, 1/500; F4/80, 1/100; Ki67, 1/1000; Ly6G, 1/60,000). The sections were rinsed with wash buffer and appropriate secondary antibody applied for 30 min. Sections for CD3, had Rabbit EnVision applied, CD4, CD8, F4/80 and LY6G had Rat ImmPRESS secondary solution applied. The sections were rinsed with wash buffer, visualized using DAB and then counterstained with Haematoxylin in the Intense R kit.

For IHC analysis, stained slides were scanned on the Leica Aperio AT2 slide scanner at 20x magnification. Image analysis was performed using the Indica Labs HALO® image analysis platform (Indica Labs, v3.1.1076.363) with the CytoNuclear v2.0.9 module. Briefly, the number of staining positive cells per mm² of tissue area was calculated, where the area of the tumour/stroma border was classified as ~100 μm either side of the border and the tumour centre as >200 μm from the tumour/stroma border. For each segment 10 square regions of interest (200 μm each side) were analyzed.

## Statistical analyses

Statistical analysis of RNaseq data is described above. Statistical analysis for IHC analysis was performed with GraphPad Prism v9.0.0 (GraphPad Software) using two-tailed Welch's $t$ test. Statistical comparisons of survival data were performed using the log-rank (Mantel-Cox) test. For individual value plots, data are displayed as mean +/- standard deviation. Statistical tests and corresponding $P$ values are indicated in the figure legends and figures.

## Reporting summary

Further information on research design is available in the Nature Portfolio Reporting Summary linked to this article.

# Data availability

Murine transcriptomic data are publicly available through the Gene Expression Omnibus (GEO) with the accession code GSE199070 and human transcriptomic datasets are publicly available through the NCBI BioProject (ID PRNJA844527). The remaining data are available within the Article, Supplementary Information or source data file. Source data are provided with this paper.

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

## Acknowledgements

We thank the Core Services and Advanced Technologies at the Cancer Research UK Beatson Institute (A31287 and C596/A17196) and particularly the Biological Services Unit, Histology Service and Molecular Technologies. G.J.I and members of his laboratory were supported by Cancer Research UK (A29802). O.J.S and members of his laboratory were supported by Cancer Research UK (A21139 and DRCQQR-May21\100002). K.B was supported by Cancer Research UK (A29799). M.T-T was supported by a Cancer Research UK MB-PhD studentship, M.B was supported by a British Skin Foundation PhD studentship and D.S. and P.B were supported by from the European Union's Horizon 2020 Research and Innovation Program under the Marie Skłodowska-Curie grant agreement No 861196 designated for PRECODE.

## Author contributions

P.B. performed and designed the bioinformatics analysis with some assistance from M.T-T., M.B. and D.S. R.A.R., P.C. performed in-vivo experiments supervised by K.B. and O.J.S. U-M.B. performed human RNaseq, C.S. analysed murine IHC. W.R reviewed murine tumour pathology. K.P isolated human RNA samples. J.T collated sample clinical data. R.J. performed murine RNAseq. A.D.C., R.A.R. and P.C. analysed murine tumour incidence. E.D., A.J.S., S.T.A., C.M.P., C.A.H. provided human samples. C.M.P., C.A.H., O.J.S., I.M.L., and G.J.I conceived and designed the study and obtained funding. P.B. and G.J.I. wrote the manuscript and all authors reviewed the manuscript.

## Competing interests

The authors declare no competing interests.

## Additional information

[1]School of Cancer Sciences, University of Glasgow, Glasgow G61 1QH, UK. [2]Department of Surgery, University of Heidelberg, Heidelberg 69120, Germany. [3]Section Surgical Research, University Clinic Heidelberg, Heidelberg 69120, Germany. [4]Cancer Research UK Beatson Institute, Glasgow G61 1BD, UK. [5]Edinburgh Medical School, University of Edinburgh, Edinburgh EH16 4TJ, UK. [6]Faculty of Medicine and Dentistry, Queen Mary University of London, London E1 1BB, UK. [7]Department of Dermatology, Royal London Hospital, Barts Health NHS Trust, London E1 1BB, UK. [8]St John's Institute of Dermatology, St Thomas's Hospital, London SE1 7EP, UK. [9]1st Department of Dermatology and Venereology, Andreas Sygros Hospital, Medical School, National and Kapodistrian University of Athens, Athens 16121, Greece. [10]Department of Dermatology, University of of California at San Francisco, San Francisco, CA, USA. [11]Molecular and Clinical Medicine, School of Medicine, University of Dundee, Dundee DD1 4HN, UK. [12]Present address: Cancer Research UK Scotland Centre, Institute of Genetics and Cancer, University of Edinburgh, Edinburgh EH4 2XR, UK. [13]Present address: German Cancer Research Centre (DKFZ), Heidelberg 61920, Germany. ✉e-mail: Peter.Bailey.2@glasgow.ac.uk; i.m.leigh@qmul.ac.uk; Gareth.Inman@glasgow.ac.uk

