## [Peer Review File · Nature Communications]

REVIEWER COMMENTS

Reviewer #1 (Remarks to the Author): clinical expertise in actinic keratosis

The manuscript entitled "Driver gene combinations dictate cutaneous squamous cell carcinoma disease continuum progression" by the group of Dr. Inman uses high-resolution transcriptional profiling of patient samples of normal skin, actinic keratoses, and cutaneous squamous cell carcinoma of different clinical stages. This is a fascinating study with a potentially high significance. It defines the development of cSCC as a disease continuum differentiated into a progenitor-like state. Such a finding provides an intriguing insight into the progression of one of the most common human cancers and explains the often puzzling biology of these lesions. For instance, it is well established that tumor promoters in various other tissues (such as the entire Notch signaling pathway) act as bona fide tumor suppressors in the skin. The explanation is likely because these molecules are also drivers of keratinocyte differentiation in the epidermis. From this point of view, one could even hypothesize that the findings presented in this study are also applicable for other squamous epithelia and cancer that originate there, such as squamous cell lung cancer and head and neck SCCs.

At the same time, several weaknesses are diminishing the enthusiasm for these studies and need to be addressed before the manuscript is considered for publication in Nature Communication.

1. Origin of samples: a weakness of the study is the lack of analysis for matched samples: adjacent normal skin versus AK and adjacent normal skin vs. SCC. It is also unclear if the "normal skin" samples used in the study are from sun-exposed skin. More importantly, the authors did not provide any data on the location of the obtained samples. The whole section on a human subject in the supplement document is missing (patient characteristics, gender and age info, etc.). Without considering this info, it is hard to fully estimate the obtained findings' significance.

2. The authors will need to describe better the designations "early differentiation" and "late differentiation." For instance, the two early spinous cell proteins, keratins 1 and 10 (K1 and K10), are widely accepted as early differentiation markers and instead, they appear to be classified as the "late differentiation" signature (Suppl. Data 1).

3. The authors do not discuss some key findings of the analysis. For instance, it is quite puzzling why the population of basal tumor-specific keratinocytes (Fig. 1f) seems to be present almost without a change in all studied samples. What is the consequence of this finding for our understanding of the biology of cSCC development?

4. A significant weakness of the study is the use of the described animal models. It is unclear why the authors target the genes of interest to Lgr5+ cells only. In the skin, Lgr5 has been genetically linked to a network of genes explicitly expressed in the hair follicle (Jaks et al., Nature Genetics 2008). If anything, Lgr6+ cells are believed to be involved much more in cSCC biology.

Reviewer #2 (Remarks to the Author): expertise in skin cancer progression and evolution

The authors have created a beautiful database of expression and genetic data gathered from many human cSCC samples at various stages of progression. From these data they show that progression starts with suppression of differentiation and ends with upregulation of progenitor fate. They use their genetic analysis to uncover the breadth of mutations associated with cSCC, and follow up by genetic manipulation in murine epidermis. With this model they define the combinations of oncogenic stimuli required to initiate cSCC and then profile the transcriptome to compare to human SCC progression. Together the data are nice and represent a valuable resource, but also do not provide many significant new insights. For instance, it is no surprise that tumor progression coincides with a decrease of differentiation and increase of progenitor fate, this has been observed in most human and murine models of various cancers. The fact that human and mouse show similar transcriptional patterns is also not surprising. Perhaps the authors could

attempt to get more from their data to ask more probing questions. For instance:

1, The consensus amongst human skin biologists and dermatologists is that cSCC is initiated from the interfollicular epidermis, while in mouse there are newer data suggesting that cSCC can arise from hair follicle stem cells. Could the authors mine their data for evidence of IFE vs HFSC fate to make insights as to where these tumors formed? Or perhaps use markers that emerge from these data to look at fate in situ in adjacent tissue from where they derived tissue for profiling?

2, Related to the first issue, the authors used Lgr5Cre to initiate tumors in the mouse model, but this should limit initiation to HFSCs or at a minimum the follicle in general. The authors should at a minimum address this experimental choice. If the human tumors start from IFE and their murine model uses HFSCs as initiators, yet they still yield similar patterns. Does this mean that fate is determined by the oncogenic stimuli as opposed to the cell of origin?

3, Of course it would have been preferable to perform scRNA-seq on all these samples as opposed to bulk-rna-seq, but also cost prohibitive. Instead, some have made use of deconvolution in an attempt to understand the cellular heterogeneity of their sample, perhaps the authors could take advantage of the same? The authors managed to do this somewhat for cells of the immune system, but perhaps this could be expanded informatically.

4, There are essentially zero data/analysis in situ in this study. I think this is a missed opportunity. For the fate changes, it would be interesting to map them in situ to determine whether there are patterns between (for instance) the leading edge of tumors versus the inside of the tumor. In fact, one might be able to use such an analysis to define whether the tumors initiated in the IFE or the follicle.

5, Many groups have published data suggesting that various proteins play a critical role in SCC formation or progression, mostly with murine data. Can the authors describe the pattern for such genes (such as SOX2, PITX, KLF4, p63, Nr2F2, FAT1, Sox9) in their database to determine whether these previous studies hold up in human cancer?

6, Many groups are now using transcriptional data to glean metabolic information. One can now map the expression of metabolic genes and infer relative activity for various metabolic pathways. Given the supposed important of metabolic changes to tumor progression, this would seem to be a worthwhile analysis, and may yield novel insights that could be validated in situ or by metabolic profiling on similar samples.

7, The Benvenisty group pioneered an approach to glean genomic rearrangement data from transcriptome profiles. Given the rich dataset obtained here, the authors should be able to use their transcriptional data to probe for deletions/duplications across the genome. Essentially, the reads are simply mapped to the genome to look for regions of amplification of signal relative to adjacent transcripts. Such an analysis would make a significant contribution to the current study, perhaps explaining some of the transcriptional changes, or perhaps even identifying novel transcripts due to chromosomal rearrangements.

Reviewer #3 (Remarks to the Author): expertise in melanoma mouse models

RNA-seq was used to identify the gene expression profiling in SCC. The orchestrated suppression of epidermal differentiation and the induction of progenitor-like gene expression was found in SCC development. Comparative systems analysis of human SCC coupled with the generation of genetically engineered murine models revealed that combinatorial sequential inactivation of the Tgfbr2, Trp53, and Notch1 coupled with activation of Ras signaling progressively drives SCC progression. Some interesting information were provided to understand the gene expression profiles IN SCC. However, the overall story is very descriptive. Some hints and clues were provided only. Further studies are required to get some solid conclusion. Furthermore, the novelty of this story is standard.

Reviewer #4 (Remarks to the Author): expertise in skin cancer sequencing

General comments

This study affirms a role for *Tgfr2*, *Trp53*, and *Notch1* coupled with activation of Ras signalling in the progression of cSCC. This is not surprising as these genes have been shown to be important in cancer progression in many previous studies. However, other than TP53, are the other genes recurrently (and concurrently) altered in clinical cSCC specimens? The authors refer to their own study (ref #3) with 12 specimens with only 2 carrying RAS alterations and only one specimen carrying concurrent RAS, TGFBR1, TP53 NOTCH1/2 alterations. There are several published studies and publically available cSCC gene datasets that could be further interrogated and included here to determine the frequency of alterations in RAS, TGFBRs and NOTCH1 and thus to help validate the clinical relevance of this combination of genes.

Nevertheless, in the current study this combination of genes are shown to be key drivers of the geneset expression differences and morphological/histopath (?) changes seen along a differentiated to progenitor (ie well to poorly differentiated) axis. The paper strongly focusses on effects on differentiation status which is fine as it is a known risk factor in cSCC but it would be useful to correlate these to other known risk factors (eg patient tumour thickness, PNI, LVI etc and patient outcomes- see specific comment #1 below on clinical manifest information). There is also much evidence in the literature of the effect of TGF beta family genes in EMT and tumor progression towards a more invasive, high-risk phenotype. The authors should touch on the role of TGF beta in cSCC carcinogenesis and progression with respect to it being integral for maintaining proper skin homeostasis with tumor suppressive function but then having a tumor promoting function in cSCC as tumors progress. An immune suppressive function of TGF beta and EMT has also been reported.

If PD tumors associate with worse prognosis the authors should test the prognostic utility of their DP signature score to validate its clinical utility as well as for primary tumor classification. This would increase the clinical significance of their findings and increase the impact of the study.

The authors flip between the terms differentiation status (usually this is shown in figures), DP scores and progenitor-like state in the text which is somewhat confusing when deciphering the data in figures with how this is described in the results text. Some examples are given in specific comments below.

I felt that the Discussion is limited with very few citations to past, relevant literature. Authors could further discuss the implications of their data and observations, as well as talk to any limitations of the study. Eg the TGF beta and EMT query above. Ras mutations are not the only way of activating this signalling pathway. What are upstream RTKS doing in their animal model? Could also further discuss the potential roles of the different T cell subpopulations and monocyte/macrophages seen in the DP continuum – eg are these tumor promoting or not? There should be a discussion on the pros and cons on their GEMM model given that it is not a UV-induced one and, quoting the authors, that “The orchestrated sequence of events as disease progresses from a differentiated to a progenitor like state are likely to be dictated by the genetic changes caused by chronic UV damage”. Is the lack of correlation between TMB and DP axis (signature?) meant to suggest that this may not necessarily be the case given that combinatorial inactivation of *Tgfr2*, *Notch1*, *Trp53* and mutational activation of *Kras* generated cSCCs efficiently? Others have previously shown that it can be easy to generate skin tumors with only few gene inactivations/activations (including RAS) <https://doi.org/10.1038/nrc1838> but whether this is clinically relevant is unclear given the strong UV-signature in cSCC. It would be interesting to compare the findings of their GEMM model to the many UV-induced mouse model studies in the literature that model the progression from premalignant AK to cSCC and have shown similar histopathology between these murine and human cSCC as well as very similar mutational landscape and mutational signatures to human cSCC. What commonalities are there?

Specific comments

1. Other than the differentiation status the clinical manifest has very limited patient histopathological data (eg tumour thickness, PNI, LVI etc). This information should be included and some discussion around the geneset findings and relationship to DP signature included. Is the

normal skin from sun exposed areas, adjacent to tumours or for distant non-exposed sites? Or from a mix of all 3? Is the differentiation status in this table based on histopath scoring or their DP score?

2. Line 109: Extended Data Fig. 1e does show that normal skin and AK samples mostly cluster into class 1 but there are several AKs clustering well within Class 2 (cSCC and met) samples. Authors should comment on these – were these perhaps wrongly designated as AKs by histopath?

3. Line 123- 124 suggests that the genesets in figure 1a were derived from 1 (only?) published dataset but it isn't clear from the related Line 539 in figure 1 legend: "Consensus clustering identifies two classes of samples identified by RNAseq." whether they are referring to data from their study or from the literature?? This needs to be clarified and the use of data from only one organotypic model justified further.

4. Not surprising that " normal skin and AK are significantly enriched for genes associated with late epidermal differentiation, whereas primary and metastatic SCC are significantly enriched for genes associated with skin progenitor cells (Fig. 1b)." and that cSCC "represents a continuum of epidermal de-differentiation" (Fig 1c). The DP score shows this continuum reasonably well for skin v AK v tumour separation but there seems to be quite a bit of intermingling of differentiation status, esp towards the right hand side which reduces the confidence in this "DP signature score".

5. Extended data fig 2 legends needs more detail – eg are medians +/- what range shown here? The separated WD-MD (n = 2 samples) data set is not useful

6. Line 229 – extended data fig 4a - I dont see a clear separation between immune pathways and cell types esp for TKS and Tumor-KCs-diff

7. Figure 2 legend: by "Heatmap shows 9 core co-expressed gene clusters ordered by DP signature score" do the authors actually mean by differentiation status as per the legend? Please provide more information regarding this statement: "Nine of these clusters exhibited significant correlation with the Differentiation-Progenitor score (Fig. 2a)", supplementary data etc.

8. And related text (Lines 255-259): authors refer to cluster 3 as "enriched for pathways involved in formation of the cornified envelope, keratinization, neutrophil degranulation as well as antimicrobial peptides (Fig. 2a)." Should they be referring to cluster 2 not 3?? In any case, it looks like several other clusters either precede/are coincident with changes in immunomodulatory processes and/or other non-immuno processes. Anyway, it is not at all clear how the authors arrived at their conclusions in reference to Figure 2a,b.

Bioinformatics comments

1. Clarify whether the statements in the following two lines refer to data selected from the PCA plot (top 2k variable genes, Extended Data Figure 1)

Line 162: To decipher the molecular pathways and/or processes underpinning SCC progression we performed k-means clustering using the 2000 most significantly differentially expressed genes between clinical designations.

Line 687: Hierarchical k-means clustering was performed using normalised gene expression representing the 2000 most differentially expressed genes between the differentiated and progenitor-like states.

2. Supp file 2; Please provide full GSEA result Tables and which annotation type was used (GO/Reactome) and list the query genes or number of genes (In the method section it only says; "using genes significantly and differentially expressed").

Excel entries shows Reactome, whereas tab name reads GO?

RESPONSE TO REVIEWERS' COMMENTS

Reviewer #1 (Remarks to the Author): clinical expertise in actinic keratosis

The manuscript entitled "Driver gene combinations dictate cutaneous squamous cell carcinoma disease continuum progression" by the group of Dr. Inman uses high-resolution transcriptional profiling of patient samples of normal skin, actinic keratoses, and cutaneous squamous cell carcinoma of different clinical stages. This is a fascinating study with a potentially high significance. It defines the development of cSCC as a disease continuum differentiated into a progenitor-like state. Such a finding provides an intriguing insight into the progression of one of the most common human cancers and explains the often puzzling biology of these lesions. For instance, it is well established that tumor promoters in various other tissues (such as the entire Notch signaling pathway) act as bona fide tumor suppressors in the skin. The explanation is likely because these molecules are also drivers of keratinocyte differentiation in the epidermis. From this point of view, one could even hypothesize that the findings presented in this study are also applicable for other squamous epithelia and cancer that originate there, such as squamous cell lung cancer and head and neck SCCs.

At the same time, several weaknesses are diminishing the enthusiasm for these studies and need to be addressed before the manuscript is considered for publication in Nature Communication.

1. Origin of samples: a weakness of the study is the lack of analysis for matched samples: adjacent normal skin versus AK and adjacent normal skin vs. SCC. It is also unclear if the "normal skin" samples used in the study are from sun-exposed skin. More importantly, the authors did not provide any data on the location of the obtained samples. The whole section on a human subject in the supplement document is missing (patient characteristics, gender and age info, etc.). Without considering this info, it is hard to fully estimate the obtained findings' significance.

2. The authors will need to describe better the designations "early differentiation" and "late differentiation." For instance, the two early spinous cell proteins, keratins 1 and 10 (K1 and K10), are widely accepted as early differentiation markers and instead, they appear to be classified as the "late differentiation" signature (Suppl. Data 1).

3. The authors do not discuss some key findings of the analysis. For instance, it is quite puzzling why the population of basal tumor-specific keratinocytes (Fig. 1f) seems to be present almost without a change in all studied samples. What is the consequence of this finding for our understanding of the biology of cSCC development?

4. A significant weakness of the study is the use of the described animal models. It is unclear why the authors target the genes of interest to Lgr5+ cells only. In the skin, Lgr5 has been genetically linked to a network of genes explicitly expressed in the hair follicle (Jaks et al., Nature Genetics 2008). If anything, Lgr6+ cells are believed to be involved much more in cSCC biology.

Reviewer #2 (Remarks to the Author): expertise in skin cancer progression and evolution

The authors have created a beautiful database of expression and genetic data gathered from many human cSCC samples at various stages of progression. From these data they show that progression starts with suppression of differentiation and ends with upregulation of progenitor fate. They use their genetic analysis to uncover the breadth of mutations associated with cSCC, and follow up by genetic manipulation in murine epidermis. With this model they define the combinations of oncogenic stimuli required to initiate cSCC and then profile the transcriptome to compare to human SCC progression. Together the data are nice and represent a valuable resource, but also do not provide many significant new insights. For instance, it no surprise that tumor progression coincides with a decrease of differentiation and increase of progenitor fate, this has been observed in most human and murine models of various cancers. The fact that human and mouse show similar transcriptional patterns is also not surprising. Perhaps the authors could attempt to get more from their data to ask more probing questions. For instance:

1, The consensus amongst human skin biologists and dermatologists is that cSCC is initiated from the interfollicular epidermis, while in mouse there are newer data suggesting that cSCC can arise from hair follicle stem cells. Could the authors mice their data for evidence of IFE vs HFSC fate to make insights as to where these tumors formed? Or perhaps use markers that emerge from these data to look at fate in situ in adjacent tissue from where they derived tissue for profiling?

2, Related to the first issue, the authors used Lgr5Cre to initiate tumors in the mouse model, but this should limit initiation to HFSCs or at a minimum the follicle in general. The authors should at a minimum address this experimental choice. If the human tumors start from IFE and their murine model uses HFSCs as initiators, yet they still yield similar patterns. Does this mean that fate is determined by the oncogenic stimuli as opposed to the cell of origin?

3, Of course it would have been preferable to perform scRNA-seq on all these samples as opposed to bulk-rna-seq, but also cost prohibitive. Instead, some have made use of deconvolution in an attempt to understand the cellular heterogeneity of their sample, perhaps the authors could take advantage of the same? The authors managed to do this somewhat for cells of the immune system, but perhaps this could be expanded informatically.

4, There are essentially zero data/analysis in situ in this study. I think this is a missed opportunity. For the fate changes, it would be interesting to map them in situ to determine whether there are patterns between (for instance) the leading edge of tumors versus the inside of the tumor. In fact, one might be able to use such an analysis to define whether the tumors initiated in the IFE or the follicle.

5, Many groups have published data suggesting that various proteins play a critical role in SCC formation or progression, mostly with murine data. Can the authors describe the pattern for such genes (such as SOX2, PITX, KLF4, p63, Nr2F2, FAT1, Sox9) in their database

to determine whether these previous studies hold up in human cancer?

6, Many groups are now using transcriptional data to glean metabolic information. One can now map the expression of metabolic genes and infer relative activity for various metabolic pathways. Given the supposed importance of metabolic changes to tumor progression, this would seem to be a worthwhile analysis, and may yield novel insights that could be validated *in situ* or by metabolic profiling on similar samples.

7, The Benvenisty group pioneered an approach to glean genomic rearrangement data from transcriptome profiles. Given the rich dataset obtained here, the authors should be able to use their transcriptional data to probe for deletions/duplications across the genome. Essentially, the reads are simply mapped to the genome to look for regions of amplification of signal relative to adjacent transcripts. Such an analysis would make a significant contribution to the current study, perhaps explaining some of the transcriptional changes, or perhaps even identifying novel transcripts due to chromosomal rearrangements.

Reviewer #3 (Remarks to the Author): expertise in melanoma mouse models

RNA-seq was used to identify the gene expression profiling in SCC. The orchestrated suppression of epidermal differentiation and the induction of progenitor-like gene expression was found in SCC development. Comparative systems analysis of human SCC coupled with the generation of genetically engineered murine models revealed that combinatorial sequential inactivation of the *Tgfr2*, *Trp53*, and *Notch1* coupled with activation of Ras signaling progressively drives SCC progression. Some interesting information was provided to understand the gene expression profiles in SCC. However, the overall story is very descriptive. Some hints and clues were provided only. Further studies are required to get some solid conclusion. Furthermore, the novelty of this story is standard.

Reviewer #4 (Remarks to the Author): expertise in skin cancer sequencing

General comments

This study affirms a role for *Tgfr2*, *Trp53*, and *Notch1* coupled with activation of Ras signaling in the progression of cSCC. This is not surprising as these genes have been shown to be important in cancer progression in many previous studies. However, other than TP53, are the other genes recurrently (and concurrently) altered in clinical cSCC specimens? The authors refer to their own study (ref #3) with 12 specimens with only 2 carrying RAS alterations and only one specimen carrying concurrent RAS, TGFBR1, TP53 NOTCH1/2 alterations. There are several published studies and publically available cSCC gene datasets that could be further interrogated and included here to determine the frequency of alterations in RAS, TGFBRs and NOTCH1 and thus to help validate the clinical relevance of this combination of genes.

Nevertheless, in the current study this combination of genes are shown to be key drivers of the geneset expression differences and morphological/histopath (?) changes seen along a differentiated to progenitor (ie well to poorly differentiated) axis. The paper strongly focuses on effects on differentiation status which is fine as it is a known risk factor in cSCC

but it would be useful to correlate these to other known risk factors (eg patient tumour thickness, PNI, LVI etc and patient outcomes- see specific comment #1 below on clinical manifest information). There is also much evidence in the literature of the effect of TGF beta family genes in EMT and tumor progression towards a more invasive, high-risk phenotype. The authors should touch on the role of TGF beta in cSCC carcinogenesis and progression with respect to it being integral for maintaining proper skin homeostasis with tumor suppressive function but then having a tumor promoting function in cSCC as tumors progress. An immune suppressive function of TGF beta and EMT has also been reported.

If PD tumors associate with worse prognosis the authors should test the prognostic utility of their DP signature score to validate its clinical utility as well as for primary tumor classification. This would increase the clinical significance of their findings and increase the impact of the study.

The authors flip between the terms differentiation status (usually this is shown in figures), DP scores and progenitor-like state in the text which is somewhat confusing when deciphering the data in figures with how this is described in the results text. Some examples are given in specific comments below.

I felt that the Discussion is limited with very few citations to past, relevant literature. Authors could further discuss the implications of their data and observations, as well as talk to any limitations of the study. Eg the TGF beta and EMT query above. Ras mutations are not the only way of activating this signalling pathway. What are upstream RTKS doing in their animal model? Could also further discuss the potential roles of the different T cell subpopulations and monocyte/macrophages seen in the DP continuum – eg are these tumor promoting or not?

There should be a discussion on the pros and cons on their GEMM model given that it is not a UV-induced one and, quoting the authors, that “The orchestrated sequence of events as disease progresses from a differentiated to a progenitor like state are likely to be dictated by the genetic changes caused by chronic UV damage”. Is the lack of correlation between TMB and DP axis (signature?) meant to suggest that this may not necessarily be the case given that combinatorial inactivation of Tgfbr2, Notch1, Trp53 and mutational activation of Kras generated cSCCs efficiently? Others have previously shown that it can be easy to generate skin tumors with only few gene inactivations/activations (including RAS) <https://doi.org/10.1038/nrc1838> but whether this is clinically relevant is unclear given the strong UV-signature in cSCC. It would be interesting to compare the findings of their GEMM model to the many UV-induced mouse model studies in the literature that model the progression from premalignant AK to cSCC and have shown similar histopathology between these murine and human cSCC as well as very similar mutational landscape and mutational signatures to human cSCC. What commonalities are there?

Specific comments

1. Other than the differentiation status the clinical manifest has very limited patient histopathological data (eg tumour thickness, PNI, LVI etc). This information should be included and some discussion around the geneset findings and relationship to DP signature

included. Is the normal skin from sun exposed areas, adjacent to tumours or for distant non-exposed sites? Or from a mix of all 3? Is the differentiation status in this table based on histopath scoring or their DP score?

2. Line 109: Extended Data Fig. 1e does show that normal skin and AK samples mostly cluster into class 1 but there are several AKs clustering well within Class 2 (cSCC and met) samples. Authors should comment on these – were these perhaps wrongly designated as AKs by histopath?

3. Line 123- 124 suggests that the genesets in figure 1a were derived from 1 (only?) published dataset but it isn't clear from the related Line 539 in figure 1 legend: "Consensus clustering identifies two classes of samples identified by RNAseq." whether they are referring to data from their study or from the literature?? This needs to be clarified and the use of data from only one organotypic model justified further.

4. Not surprising that "normal skin and AK are significantly enriched for genes associated with late epidermal differentiation, whereas primary and metastatic SCC are significantly enriched for genes associated with skin progenitor cells (Fig. 1b)." and that cSCC "represents a continuum of epidermal de-differentiation" (Fig 1c). The DP score shows this continuum reasonably well for skin v AK v tumour separation but there seems to be quite a bit of intermingling of differentiation status, esp towards the right hand side which reduces the confidence in this "DP signature score".

5. Extended data fig 2 legends needs more detail – eg are medians +/- what range shown here? The separated WD-MD (n = 2 samples) data set is not useful

6. Line 229 – extended data fig 4a - I dont see a clear separation between immune pathways and cell types esp for TKS and Tumor-KCs-diff

7. Figure 2 legend: by "Heatmap shows 9 core co-expressed gene clusters ordered by DP signature score" do the authors actually mean by differentiation status as per the legend? Please provide more information regarding this statement: "Nine of these clusters exhibited significant correlation with the Differentiation-Progenitor score (Fig. 2a)", supplementary data etc.

8. And related text (Lines 255-259): authors refer to cluster 3 as "enriched for pathways involved in formation of the cornified envelope, keratinization, neutrophil degranulation as well as antimicrobial peptides (Fig. 2a)." Should they be referring to cluster 2 not 3?? In any case, it looks like several other clusters either precede/are coincident with changes in immunomodulatory processes and/or other non-immuno processes. Anyway, it is not at all clear how the authors arrived at their conclusions in reference to Figure 2a,b.

Bioinformatics comments

1. Clarify whether the statements in the following two lines refer to data selected from the PCA plot (top 2k variable genes, Extended Data Figure 1)

Line 162: To decipher the molecular pathways and/or processes underpinning SCC progression we performed k-means clustering using the 2000 most significantly differentially expressed genes between clinical designations.

Line 687: Hierarchical k-means clustering was performed using normalised gene expression representing the 2000 most differentially expressed genes between the differentiated and progenitor-like states.

2. Supp file 2; Please provide full GSEA result Tables and which annotation type was used (GO/Reactome) and list the query genes or number of genes (In the method section it only says; "using genes significantly and differentially expressed").

Excel entries shows Reactome, whereas tab name reads GO?

Response to reviewers.

Reviewer #1: clinical expertise in actinic keratosis

Reviewer #2: expertise in skin cancer progression and evolution

Reviewer #3: expertise in melanoma mouse models

Reviewer #4: expertise in skin cancer sequencing

We are grateful to the reviewers for their time spent reviewing our manuscript and for their mainly positive comments and constructive criticisms. We have taken on board their comments and revised our manuscript accordingly. Below, we have provided a point-by-point response to the reviewers' comments, with their comments in black print and our responses highlighted in blue. In addition, we have included two figures especially for the review process. We believe that this has greatly improved our manuscript and added to its novelty and importance.

Reviewer #1 (Remarks to the Author):

The manuscript entitled "Driver gene combinations dictate cutaneous squamous cell carcinoma disease continuum progression" by the group of Dr. Inman uses high-resolution transcriptional profiling of patient samples of normal skin, actinic keratoses, and cutaneous squamous cell carcinoma of different clinical stages. This is a fascinating study with a potentially high significance. It defines the development of cSCC as a disease continuum differentiated into a progenitor-like state. Such a finding provides an intriguing insight into the progression of one of the most common human cancers and explains the often puzzling biology of these lesions. For instance, it is well established that tumor promoters in various other tissues (such as the entire Notch signaling pathway) act as bona fide tumor suppressors in the skin. The explanation is likely because these molecules are also drivers of keratinocyte differentiation in the epidermis. From this point of view, one could even hypothesize that the findings presented in this study are also applicable for other squamous epithelia and cancer that originate there, such as squamous cell lung cancer and head and neck SCCs. At the same time, several weaknesses are diminishing the enthusiasm for these studies and need to be addressed before the manuscript is considered for publication in Nature Communication.

1. Origin of samples: a weakness of the study is the lack of analysis for matched samples: adjacent normal skin versus AK and adjacent normal skin vs. SCC. It is also unclear if the "normal skin" samples used in the study are from sun-exposed skin. More importantly, the authors did not provide any data on the location of the obtained samples. The whole section on a human subject in the supplement document is missing (patient characteristics, gender and age info, etc.). Without considering this info, it is hard to fully estimate the obtained findings' significance.

We apologise for the omissions in our description of patient and tumour characteristics in our original submission and thank the reviewer for requesting this information. We have updated the clinical manifest file (within Supplementary Data File 1) to include Age, Sex, tumour site, tumour depth, tumour diameter, and invasion parameters described in the clinical pathology reports and included labelling of our normal skin samples which are all taken from sun exposed skin with some of these samples being peri-lesional skin. Addition of this information has now enabled us to compare DvP signature scores not just with differentiation status but also with tumour depth, tumour diameter, invasion status, Age and Sex (New Supplementary Figure 4d-f). Importantly tumour depth and diameter significantly correlate with progenitor like status whereas age and sex do not. We were only able to retrieve definitive invasion status data (beyond fat or perineural or lympho-vascular) from the clinical records of 7 of our samples. These had a tendency to be more progenitor like but this failed to reach statistical significance. Taken together our analyses indicate that progenitor like status correlates with tumour size, depth and pathological differentiation status further supporting our classification of disease progression and we hope the reviewer agrees that this now adds further to the significance of our findings.

We thank the reviewer for suggesting matched sample analysis. We have described the relationship of our samples in a new Venn diagram (New Supplementary Figure 2a). This illustrated that our sample set contains 25 matched primary tumours and normal sun exposed skin samples taken from the same patients. We have only 1 matched primary tumour and AK and no normal sun exposed skin matched to our AK samples. Consequently, we have performed matched differential gene expression analysis between the matched 25 pairs of normal sun exposed skin and primary tumours and compared these results to the differential gene expression analysis of all normal (n=26) and all primary tumours (n=66) (New supplementary Data file 3, New Supplementary Figure 2). Whilst there is considerable overlap of the differentially expressed genes and GSEA identified pathways and processes between the total sample and matched sample analyses there are also many unique differentially expressed genes identified in the matched analysis with GSEA highlighting the potential importance of muscle contraction and neuronal processes in disease formation. There are also many unique differentially expressed genes identified in the total sample analysis highlighting the power of both approaches to reveal key genes and processes in cSCC formation. Furthermore, we believe that consideration of the matched samples in relation to the DvP axis further exemplifies the utility of representing cSCC disease progression as a continuum as matched samples of primary tumours and normal skin are distributed across the DvP axis (New Supplementary Fig. 5a) and despite having overall significant differences in DvP signature score and expression of LOR as an example signature gene (Supplementary Fig. 5b, c) individual pairs of samples may show similar signature scores or LOR expression. We have updated the results section to include description and discussion of the matched sample analysis accordingly.

2. The authors will need to describe better the designations "early differentiation" and "late differentiation." For instance, the two early spinous cell proteins, keratins 1 and 10 (K1 and K10), are widely accepted as early differentiation markers and instead, they appear to be classified as the "late differentiation" signature (Suppl. Data 1).

We apologise for our lack of clarity around differentiation designations. The progenitor, early and late differentiation gene signatures were defined by Paul A. Khavari's group (Lopez-Pajares, V., et al., A LncRNA-MAF:MAFB transcription factor network regulates epidermal differentiation. *Dev Cell*, 2015. 32(6): p. 693-706.) using organotypic models of epidermal differentiation. Model generation involved seeding progenitor keratinocytes on native human dermal mesenchymal tissue and performing a 7-day time course in a process that culminated in the production of a fully stratified epithelium, expressing both early and late differentiation markers. Representative differentiation markers for the spinous layer (including keratin 1 and Keratin 10) first appeared on day 3, whereas markers for the outer granular layer of the skin, such as filaggrin and loricrin, were detected later in the time course at days 4 and 5, respectively. This regeneration time course therefore recapitulated the earlier and later induction of differentiation markers characteristic of the epidermis. In their classification of early and late differentiation signatures the Khavari lab included the spinous and later granular differentiation genes in their "late differentiation" signature. We have retained the original naming of these gene signatures and reference them in the amended methods section - see line 730 and provide them in Supplementary Data File 5. We also demonstrate that spinous K1 and K10 expression is lost in primary tumours that have a progenitor-like transcriptomic profile (loss of late and early differentiation signatures). This is consistent with our stated conclusion that increased progenitor-like transcriptional states are associated with the loss of late/early skin differentiation programmes. We have amended the results section of our manuscript to now state "The delineation of TFs according to differentiation status closely mirrored the selective expression of keratins (K) including K1, K10, K5 and K14 (Supplementary Fig. 9a). K1 and K10 genes are present in the late differentiation gene signature (Supplementary Data File 5) and their expression demarcates the intermediated spinous layer whereas K5 and K14 are expressed in the basal layer" (Lines 229-233). We hope this now clarifies the original ambiguity.

2. The authors do not discuss some key findings of the analysis. For instance, it is quite puzzling why the population of basal tumor-specific keratinocytes (Fig. 1f) seems to be present almost without a change in all studied samples. What is the consequence of this finding for our understanding of the biology of cSCC development?

We thank the reviewer for their careful scrutiny of our CibersortX analysis and we have now presented the full CibersortX profiles of all 7 keratinocyte populations in new Supplementary Figure 6. We have added the following statement to the results section "Interestingly we observed a modulation of the tumour keratinocyte cycling population across the continuum, the appearance of the normal keratinocyte cycling population in progenitor like samples and no notable changes in basal keratinocyte populations (Supplementary Fig. 6) and these observations warrant further investigation" to highlight the reviewers point and illustrate that we do not yet know how these findings relate to our understanding of the biology of cSCC development.

4. A significant weakness of the study is the use of the described animal models. It is unclear why the authors target the genes of interest to Lgr5+ cells only. In the skin, Lgr5 has been genetically linked to a network of genes explicitly expressed in the hair follicle (Jaks et al., *Nature Genetics* 2008). If anything, Lgr6+ cells are believed to be involved much more in cSCC biology.

We chose to target our driver genes of interest to the Lgr5+ve stem cells in the murine skin as proof of principle experiments to test if, as suggested by our human analyses, that

despite considerable genetic complexity it is the combinations of driver genes that are responsible for disease progression in cSCC. We and others have previously demonstrated that murine SCCs that phenotypically resemble human cSCC can efficiently initiate from Lgr5+ve cells (Cammareri et al., 2016; Latil et al., 2017) and that cSCCs originating from these cells can be invasive, aggressive and exhibit EMT phenotypes (Latil et al., 2017; Mauri et al., 2021; Pastushenko et al., 2018, 2021). Therefore, targeting our genetic events to the Lgr5+ve stem cells in the skin should enable us to potentially study the entire disease continuum from early to advanced disease (we have added a sentence to the results section (Line 362-365) which we hope will now make our choice of models clear). More importantly, this is indeed what we found to be the case. Our LPT tumours represented very early human disease, addition of the loss of Notch1 (LNPT) resulted in tumours that transcriptionally mimicked human tumours from the middle regions of the disease continuum whereas loss of TGFBR2 coupled with mutational activation of KRAS resulted in tumours transcriptionally aligned with tumours from the middle to the end of the DvP continuum. Furthermore, our murine tumours recapitulated the coordinated modulation of transcription factors, changes in the EDC and the immune landscape that we observed in our human tumour samples. These observations, we strongly argue, demonstrate the strengths of our chosen murine models which have indeed enabled us to demonstrate that it is the driver gene combinations that dictate disease progression. In addition, in response to a point raised by Reviewer 4 we have also compared the transcriptional landscape of our genetically induced murine tumours to samples transcriptionally profiled from UV induced skin carcinogenesis in the hairless mouse (Chitsazzadeh et al., 2016). This analysis reveals a remarkable conservation of disease progression between samples (new Supplementary Figure 17) and further supports the models used within our study.

Reviewer #2 (Remarks to the Author):

The authors have created a beautiful database of expression and genetic data gathered from many human cSCC samples at various stages of progression. From these data they show that progression starts with suppression of differentiation and ends with upregulation of progenitor fate. They use their genetic analysis to uncover the breadth of mutations associated with cSCC, and follow up by genetic manipulation in murine epidermis. With this model they define the combinations of oncogenic stimuli required to initiate cSCC and then profile the transcriptome to compare to human SCC progression. Together the data are nice and represent a valuable resource, but also do not provide many significant new insights. For instance, it no surprise that tumor progression coincides with a decrease of differentiation and increase of progenitor fate, this has been observed in most human and murine models of various cancers. The fact that human and mouse show similar transcriptional patterns is also not surprising. Perhaps the authors could attempt to get more from their data to ask more probing questions. For instance:

- 1, The consensus amongst human skin biologists and dermatologists is that cSCC is initiated from the interfollicular epidermis, while in mouse there are newer data suggesting that cSCC can arise from hair follicle stem cells. Could the authors mine their data for evidence of IFE vs HFSC fate to make insights as to where these tumors formed? Or perhaps use

markers that emerge from these data to look at fate in situ in adjacent tissue from where they derived tissue for profiling?

We thank the reviewer for this excellent suggestion. We used deconvolution analysis and hair follicle cell state transcriptional signatures representing 23 cell types/states (Takahashi et al., 2020) to mine our human data for potential indicators of cell of origin (New Supplementary Figure 18, New Supplementary Data file 17). This analysis revealed increases in IFE basal, IFE Mitotic, lower bulge and outer root sheath suprabasal signatures during disease progression. The latter 2 cell types are derived from Lgr5 positive cell compartments and excitingly suggest a possible contribution of these cells as well as cells of the IFE to progenitor like human tumours. We have added a new paragraph to the results section (lines 449-475) describing these findings in detail and a new sentence to the discussion. We agree with the reviewer that it could be very informative to look at markers of these cell states in-situ but unfortunately we do not have any available material from the samples we profiled in our study to do so and we will endeavour in future prospective studies to do so.

2, Related to the first issue, the authors used Lgr5Cre to initiate tumors in the mouse model, but this should limit initiation to HFSCs or at a minimum the follicle in general. The authors should at a minimum address this experimental choice. If the human tumors start from IFE and their murine model uses HFSCs as initiators, yet they still yield similar patterns. Does this mean that fate is determined by the oncogenic stimuli as opposed to the cell of origin?

In response to a similar point raised by reviewer 1 (reviewer 1 point 4) we have further addressed our use of the Lgr5Cre to target genetic events to Lgr5+ve hair follicle stem cells in the murine skin. We have restated our above response again here: "We chose to target our driver genes of interest to the Lgr5+ve stem cells in the murine skin as proof of principle experiments to test if, as suggested by our human analyses, that despite considerable genetic complexity it is the combinations of driver genes that are responsible for disease progression in cSCC. We and others have previously demonstrated that murine SCCs that phenotypically resemble human cSCC can efficiently initiate from Lgr5+ve cells (Cammareri et al., 2016; Latil et al., 2017) and that cSCCs originating from these cells can be invasive, aggressive and exhibit EMT phenotypes (Latil et al., 2017; Mauri et al., 2021; Pastushenko et al., 2018, 2021). Therefore, targeting our genetic events to the Lgr5+ve stem cells in the skin should enable us to potentially study the entire disease continuum from early to advanced disease (we have added a sentence to the results section (Lines 362-365) which we hope will now make our choice of models clear). More importantly, this is indeed what we found to be the case. Our LPT tumours represented very early human disease, addition of the loss of Notch1 (LNPT) resulted in tumours that transcriptionally mimicked human tumours from the middle regions of the disease continuum whereas loss of TGFBR2 coupled with mutational activation of KRAS resulted in tumours transcriptionally aligned with tumours from the middle to the end of the DvP continuum. Furthermore, our murine tumours recapitulated the coordinated modulation of transcription factors, changes in the EDC and the immune landscape that we observed in our human tumour samples. These observations, we strongly argue,

demonstrate the strengths of our chosen murine models which have indeed enabled us to demonstrate that it is the driver gene combinations that dictate disease progression. In addition, in response to a point raised by Reviewer 4 we have also compared the transcriptional landscape of our genetically induced murine tumours to samples transcriptionally profiled from UV induced skin carcinogenesis in the hairless mouse (Chitsazzadeh et al., 2016). This analysis reveals a remarkable conservation of disease progression between samples (new Supplementary Figure 17) and further supports the models used within our study.

The reviewer poses the additional insightful comment, “Does this mean that fate is determined by the oncogenic stimuli as opposed to the cell of origin?”. Our deconvolution analysis of the human samples using hair follicle cell state signatures indicates that our more progenitor like samples contain both IFE and Lgr5+ve cell compartment signatures indicating potential cells of origin from both of these cell compartments making a definitive answer to this question difficult. We do however believe that the current best interpretation of our findings is “that regardless of the mechanism that generates driver gene alterations or the cells in which they take place it is the driver gene combinations that dictate cSCC disease progression” and we have added this statement to our results section (Lines 473-475).

3, Of course it would have been preferable to perform scRNA-seq on all these samples as opposed to bulk-rna-seq, but also cost prohibitive. Instead, some have made use of deconvolution in an attempt to understand the cellular heterogeneity of their sample, perhaps the authors could take advantage of the same? The authors managed to do this somewhat for cells of the immune system, but perhaps this could be expanded informatically.

We thank the reviewer for suggesting further deconvolution analyses to glean more information from our bulk RNaseq analyses as also suggested in their point 1 above. Our manuscript now contains three separate deconvolution analyses: 1) utilising CibersortX we have interrogated the population dynamics of the 7 keratinocyte populations described by Paul Khavari’s lab and the findings from these analyses are presented in Figure 1d-g, Figure 3d, Figure 4e and new Supplementary Figure 6 and are now further discussed in the results and discussion sections. 2) Immune cell deconvolution was performed using xCELL and the results are presented in Figure 3a and Figure 5f and the findings are discussed in the results and discussion sections. 3) We used deconvolution analysis and hair follicle cell state transcriptional signatures representing 23 cell types/states (Takahashi et al., 2020) to mine our human data for potential indicators of cell of origin. The intriguing findings from this new analysis are presented in (New Supplementary Figure 18, New Supplementary Data file 17) and are described and discussed in the revised manuscript. We believe that these analyses provide considerable insight into the cell population dynamics during cSCC disease progression.

4, There are essentially zero data/analysis in situ in this study. I think this is a missed opportunity. For the fate changes, it would be interesting to map them in situ to determine whether there are patterns between (for instance) the leading edge of tumors versus the inside of the tumor. In fact, one might be able to use such an analysis to define whether the tumors initiated in the IFE or the follicle.

We agree with the reviewer that in-situ analysis would provide additional valuable information to our study but unfortunately we are unable to perform this analysis due to the lack of any

further suitable material from the patient samples analysed in this current study. We have provided further analysis into potential initiation sites of the human tumours as discussed above and we hope that the reviewer now shares the belief that our improved manuscript provides significant new insights into cSCC disease progression. These insights will also provide the platform for future studies designed to interrogate critical cell interactions which support disease progression which will hopefully lead to future effective therapeutic strategies.

5, Many groups have published data suggesting that various proteins play a critical role in SCC formation or progression, mostly with murine data. Can the authors describe the pattern for such genes (such as SOX2, PITX, KLF4, p63, Nr2F2, FAT1, Sox9) in their database to determine whether these previous studies hold up in human cancer?

We thank the reviewer for appreciating the power of our database to enable queries such as these to be easily performed. We provide the specific analyses requested above at the end of this document as Reviewer Figure 1. There appears to be an elevation in the expression of SOX2, PITX1 and TP63 and a marked downregulation of KLF4 during disease progression. SOX9 and NR2F2 show variable levels of expression and FAT1 levels show an unexpected elevation of expression given its documented tumour suppressor role.

6, Many groups are now using transcriptional data to glean metabolic information. One can now map the expression of metabolic genes and infer relative activity for various metabolic pathways. Given the supposed importance of metabolic changes to tumor progression, this would seem to be a worthwhile analysis, and may yield novel insights that could be validated in situ or by metabolic profiling on similar samples.

We thank the reviewer for highlighting this important area. We have further mined our data to reveal modulation of metabolic processes during cSCC disease progression and present these findings as new Supplementary Figures 7 and 8 and new Supplementary Data file 8 and discussed these findings in the manuscript. We have also now highlighted that our K-means clustering analysis revealed downregulation of metabolism associated processes including fatty acids, sphingolipid de novo biosynthesis, cytochrome P450 and ion transport by P-type ATPases during disease progression. Our new GSEA and KEGG pathway analysis reveals the coordinated switch in gene expression of genes involved in drug metabolism, fatty acid degradation, glycolysis and gluconeogenesis and glutathione metabolism (Supplementary Figure 7a-d). To further explore changes during the progression to a progenitor like state we performed GSEA and GO ontology analysis when comparing progenitor like samples (Quartile 1 and 2) to differentiated like samples (Quartiles 3 and 4) (Supplementary Figure 8a, Supplementary Data File 8). This also highlighted profound changes in metabolic processes with suppression of lipid metabolic processes being particularly prominent (Supplementary Fig. 8a, b). Taken together, these findings indicate a switch from fatty acid and lipid metabolism to a glycolytic like state is likely to occur coincident with the acquisition of a progenitor like state. These findings are now described in the text (lines 195-208). As discussed above it would indeed be valuable to take these observations forward in future studies but due to the limitation in the availability of further samples from the patients studied here this is unfortunately not possible in the context of the current manuscript.

7, The Benvenisty group pioneered an approach to glean genomic rearrangement data from transcriptome profiles. Given the rich dataset obtained here, the authors should be able to use their transcriptional data to probe for deletions/duplications across the genome. Essentially, the reads are simply mapped to the genome to look for regions of amplification of signal relative to adjacent transcripts. Such an analysis would make a significant contribution to the current study, perhaps explaining some of the transcriptional changes, or perhaps even identifying novel transcripts due to chromosomal rearrangements.

We thank the reviewer for another excellent suggestion. We have estimated CNVs from our bulk RNAseq data utilising CaSpER. (New Supplementary Fig 12, New Supplementary Data File 11, new results description lines 328-344). This revealed a marked increase in copy number alterations observed in more progenitor like samples with a significant increase in the percentage of the genome altered with decreasing DvP quartile (Supplementary Fig. 12b,c, Kruskal-Wallis, $p=4.5 \times 10^{-8}$). Notable copy number losses of 3p and 9p and gains of 19p were observed in both analyses (Supplementary Data File 11). 26% of genes affected by copy number loss (83 out of 315) also showed a significant down regulation of gene expression ($\text{padj} < 0.05$) when comparing Quartile 1 and Quartile 2 samples to Quartile 3 and Quartile 4 samples indicating a potential tumour suppressor role for these genes (Supplementary Data File 11). Many immunoglobulin heavy chain genes showed copy number gains and an elevation of gene expression likely reflecting enhanced B cell infiltration in progenitor like samples. Excluding these, 31% of genes (14 out of 45) showed both a copy number gain and a significant elevation of gene expression in progenitor like samples indicating potential tumour promoter roles (Supplementary Data File 11). We have also added to our discussion (Lines 521-526) to reflect on these findings. We did not observe novel transcripts generated by chromosomal arrangements using this approach.

Reviewer #3 (Remarks to the Author):

RNA-seq was used to identify the gene expression profiling in SCC. The orchestrated suppression of epidermal differentiation and the induction of progenitor-like gene expression was found in SCC development. Comparative systems analysis of human SCC coupled with the generation of genetically engineered murine models revealed that combinatorial sequential inactivation of the Tgfr2, Trp53, and Notch1 coupled with activation of Ras signaling progressively drives SCC progression. Some interesting information were provided to understand the gene expression profiles IN SCC. However, the overall story is very descriptive. Some hints and clues were provided only. Further studies are required to get some solid conclusion. Furthermore, the novelty of this story is standard.

We thank the reviewer for their comments. We would like to take the opportunity to quote the other reviewers who stated:- "This is a fascinating study with a potentially high significance. It defines the development of cSCC as a disease continuum differentiated into a progenitor-like state. Such a finding provides an intriguing insight into the progression of one of the most common human cancers and explains the often puzzling biology of these lesions", "The authors have created a beautiful database of expression and genetic data gathered from many human cSCC samples at various stages of progression", "They use their genetic analysis to uncover the breadth of mutations associated with cSCC, and follow up by genetic manipulation in murine epidermis. With this model they define the combinations of oncogenic stimuli required to initiate cSCC and then profile the transcriptome to compare to human SCC progression. Together the data are nice and represent a valuable resource". In response to other comments made by the other reviewers we have performed a considerable amount of

further analysis which has further strengthened our manuscript and we now sincerely hope that the reviewer now considers our paper to be of sufficient novelty to warrant publication.

Reviewer #4 (Remarks to the Author):

General comments

This study affirms a role for *Tgfb2*, *Trp53*, and *Notch1* coupled with activation of Ras signalling in the progression of cSCC. This is not surprising as these genes have been shown to be important in cancer progression in many previous studies. However, other than TP53, are the other genes recurrently (and concurrently) altered in clinical cSCC specimens? The authors refer to their own study (ref #3) with 12 specimens with only 2 carrying RAS alterations and only one specimen carrying concurrent RAS, TGFBR1, TP53 NOTCH1/2 alterations. There are several published studies and publically available cSCC gene datasets that could be further interrogated and included here to determine the frequency of alterations in RAS, TGFBRs and NOTCH1 and thus to help validate the clinical relevance of this combination of genes.

Nevertheless, in the current study this combination of genes are shown to be key drivers of the geneset expression differences and morphological/histopath (?) changes seen along a differentiated to progenitor (ie well to poorly differentiated) axis. The paper strongly focusses on effects on differentiation status which is fine as it is a known risk factor in cSCC but it would be useful to correlate these to other known risk factors (eg patient tumour thickness, PNI, LVI etc and patient outcomes- see specific comment #1 below on clinical manifest information). There is also much evidence in the literature of the effect of TGF beta family genes in EMT and tumor progression towards a more invasive, high-risk phenotype. The authors should touch on the role of TGF beta in cSCC carcinogenesis and progression with respect to it being integral for maintaining proper skin homeostasis with tumor suppressive function but then having a tumor promoting function in cSCC as tumors progress. An immune suppressive function of TGF beta and EMT has also been reported.

If PD tumors associate with worse prognosis the authors should test the prognostic utility of their DP signature score to validate its clinical utility as well as for primary tumor classification. This would increase the clinical significance of their findings and increase the impact of the study.

The authors flip between the terms differentiation status (usually this is shown in figures), DP scores and progenitor-like state in the text which is somewhat confusing when deciphering the data in figures with how this is described in the results text. Some examples are given in specific comments below.

I felt that the Discussion is limited with very few citations to past, relevant literature. Authors could further discuss the implications of their data and observations, as well as talk to any limitations of the study. Eg the TGF beta and EMT query above. Ras mutations are not

the only way of activating this signalling pathway. What are upstream RTKS doing in their animal model? Could also further discuss the potential roles of the different T cell subpopulations and monocyte/macrophages seen in the DP continuum – eg are these tumor promoting or not?

There should be a discussion on the pros and cons on their GEMM model given that it is not a UV-induced one and, quoting the authors, that “The orchestrated sequence of events as disease progresses from a differentiated to a progenitor like state are likely to be dictated by the genetic changes caused by chronic UV damage”. Is the lack of correlation between TMB and DP axis (signature?) meant to suggest that this may not necessarily be the case given that combinatorial inactivation of *Tgfbr2*, *Notch1*, *Trp53* and mutational activation of *Kras* generated cSCCs efficiently? Others have previously shown that it can be easy to generate skin tumors with only few gene inactivations/activations (including RAS) <https://doi.org/10.1038/nrc1838>) but whether this is clinically relevant is unclear given the strong UV-signature in cSCC. It would be interesting to compare the findings of their GEMM model to the many UV-induced mouse model studies in the literature that model the progression

from premalignant AK to cSCC and have shown similar histopathology between these murine and human cSCC as well as very similar mutational landscape and mutational signatures to human cSCC. What commonalities are there?

We would like to express our gratitude to the reviewer for their thorough, constructive and inciteful review of our manuscript. Firstly, we will address each of their general comments which are not further elaborated on in their specific comments in turn below, followed by addressing each of their specific comments.

There are several published studies and publically available cSCC gene datasets that could be further interrogated and included here to determine the frequency of alterations in RAS, TGFBRs and NOTCH1 and thus to help validate the clinical relevance of this combination of genes.

We thank the reviewer for this helpful suggestion and we have now included an oncoprint analysis of KRAS, HRAS, NRAS, TGFBR1, TGFBR2, TP53, NOTCH1 and NOTCH2 genes from 151 cSCC samples curated in Cbioportal (New Supplementary Figure 13) to match that provided from our own samples shown in Figure 4a and added the following statement based on this to the results section (lines 351-354) “Analysis of 151 cSCC tumour samples curated in Cbioportal indicated TP53 mutations in 76%, NOTCH1 in 55%, *TGFBR1* in 6%, *TGFBR2* in 8% and mutation of *KRAS*, *HRAS* and *NRAS* combined occurred in 17.6% of samples (Supplementary Fig. 13)”. We believe that this further illustrates the clinical relevance of the combination of genes we chose to study. This is also supported by our pathway analysis which indicates significant negative correlation of TGFβ, NOTCH1 and TP53 signalling signatures and positive correlation of ERK signalling signatures with DvP signature score with selective modulation of signalling components occurring with disease progression including downregulation of *TP53*, *TGFBR2* and *NOTCH1* in progenitor like samples (Supplementary Fig. 14; Supplementary Data File 12).

There is also much evidence in the literature of the effect of TGF beta family genes in EMT and tumor progression towards a more invasive, high-risk phenotype. The authors should touch on the role of TGF beta in cSCC carcinogenesis and progression with respect to it being integral for maintaining proper skin homeostasis with tumor suppressive function

but then having a tumor promoting function in cSCC as tumors progress. An immune suppressive function of TGF beta and EMT has also been reported.

We agree with the reviewers' assessment of TGF β signalling in cSCC and have added the following statements to our discussion, "Our findings also reconfirm a tumour suppressor role of TGF β signalling in cSCC[39]. Consistent with this, recent studies in organotypic cultures indicate that loss of TGF β signalling may promote keratinocyte invasion[52]. It is important to note that whilst we observe correlation with loss of some TGF β signalling pathway components with disease progression it is unlikely that this is obligate for cSCC formation and TGF β signalling may also play tumour promoting roles by promoting EMT and/or immune escape[53] and this requires further investigation in cSCC." (Lines 526-532).

If PD tumors associate with worse prognosis the authors should test the prognostic utility of their DP signature score to validate its clinical utility as well as for primary tumor classification. This would increase the clinical significance of their findings and increase the impact of the study.

This is an excellent point and one that requires further investigation in future studies. Unfortunately, we do not have follow on outcome data for the patients from whom samples were taken for this study and so validation of its clinical utility is beyond the scope of this study. We have now included other tumour factors including tumour depth and diameter which are also associated with poorer prognosis in cSCC (see response to specific comment 1) and DvP signature score does correlate with these factors as well. Taken together all of the "prognostic factors" available to us in this study do correlate with DvP signature score. We hope that our study will inspire other investigators as well as our group of collaborators to test the clinical utility of DvP signature score and that it will prove to have clinical impact not only for use in diagnosis but also for future treatment selection and patient management. We also hope that the biological insights provided in our study will lead to the future development of new treatment strategies. We hope that this reviewer will agree with Reviewer 1 whom states "This is a fascinating study with a potentially high significance. It defines the development of cSCC as a disease continuum differentiated into a progenitor-like state. Such a finding provides an intriguing insight into the progression of one of the most common human cancers and explains the often puzzling biology of these lesions", Overall we believe that our study will go on to have considerable impact not only from the concepts and biological insights we have highlighted in the manuscript but also because it provides a valuable resource for others to interrogate and shed further light on this common and important cancer.

I felt that the Discussion is limited with very few citations to past, relevant literature. Authors could further discuss the implications of their data and observations, as well as talk to any limitations of the study

We have expanded our discussion, included new relevant citations and highlighted many examples where we believe concepts and hypotheses warrant further investigation as best we can given space constraints and we hope this now meets the reviewer's approval.

Ras mutations are not the only way of activating this signalling pathway. What are upstream RTKS doing in their animal model?

We agree that Ras mutations are not the only way of activating this pathway and the strong correlation with signalling to ERK and DvP signature score (Supplementary Figure 4b) clearly illustrates the general importance of this pathway in disease progression. In answer to the specific question as to what upstream RTKS are doing in our animal model we have provided Reviewer Figure 2 at the end of this section which illustrates intriguing differences in gene expression levels of their genes in the different genotype tumours. We would like to the

opportunity to highlight that the resource provided by our study enables many other such analyses to be easily undertaken in our murine and/or human samples.

Could also further discuss the potential roles of the different T cell subpopulations and monocyte/macrophages seen in the DP continuum – eg are these tumor promoting or not?

We have added the following statement to our discussion, “The changes we observe in immune cell populations in our human samples identified by xCELL and in murine tumours by immunohistochemistry suggest a possible tumour promoting role in cSCC disease progression, but this requires future further in-depth analysis and functional interrogation”

There should be a discussion on the pros and cons on their GEMM model given that it is not a UV-induced one and, quoting the authors, that “The orchestrated sequence of events as disease progresses from a differentiated to a progenitor like state are likely to be dictated by the genetic changes caused by chronic UV damage”. Is the lack of correlation between TMB and DP axis (signature?) meant to suggest that this may not necessarily be the case given that combinatorial inactivation of *Tgfbr2*, *Notch1*, *Trp53* and mutational activation of *Kras* generated cSCCs efficiently? Others have previously shown that it can be easy to generate skin tumors with only few gene inactivations/activations (including RAS) <https://doi.org/10.1038/nrc1838>) but whether this is clinically relevant is unclear given the strong UV-signature in cSCC. It would be interesting to compare the findings of their GEMM model to the many UV-induced mouse model studies in the literature that model the progression from premalignant AK to cSCC and have shown similar histopathology between these murine and human cSCC as well as very similar mutational landscape and mutational signatures to human cSCC. What commonalities are there?

We thank the reviewer for this excellent suggestion and comments. We have now included a comparative analysis of high quality RNAseq data provided by the Tsai lab in their manuscript Chitsazzadeh et al., 2016 (new Supplementary Figure 17, new Supplementary Data File 16) and added the following to our results, “Murine tumours induced by solar UV radiation of hairless mice recapitulate the histopathological features of human cSCC disease progression and are similarly genetically complex[45]. Intersection and Uniform manifold Approximation and Projection (UMAP) of RNaseq data from a UV radiation treated hairless mouse model[46] with our genetically induced murine tumours indicates a remarkable overlap of disease trajectories (Supplementary Fig 17, Supplementary Data File 16). Our LPT tumours represent very early disease and overlap with chronic UV damaged skin (CHR), LNPT tumours span early disease from CHR to papilloma formation and LKT tumours overlap with invasive SCC in the UV model. This analysis provides further evidence that the underpinning driver gene events dictate disease progression regardless of the genetic complexity in which they take place and the causative mechanism which generates them.” (Lines 437-447). This is an important addition to our paper as it underpins not only the relevance of our murine models but also the fundamental concept that we propose that it is indeed the driver gene combinations found within the genetic complexity in human tumours driven by chronic UV exposure that dictates disease progression.

Specific comments

1. Other than the differentiation status the clinical manifest has very limited patient

histopathological data (eg tumour thickness, PNI, LVI etc). This information should be included and some discussion around the geneset findings and relationship to DP signature included. Is the normal skin from sun exposed areas, adjacent to tumours or for distant non-exposed sites? Or from a mix of all 3? Is the differentiation status in this table based on histopath scoring or their DP score?

We apologise for these important omissions that we made in our original description of the patient samples which were also pointed out by reviewer 1. We have updated the clinical manifest file (within Supplementary Data File 1) to include Age, Sex, tumour site, tumour depth, tumour diameter, and invasion parameters described in the clinical pathology reports and included labelling of our normal skin samples which are all taken from sun exposed skin with some of these samples being peri-lesional skin. We have added the information in the Supplementary Data File 1 descriptor to indicate that tumour characteristics including differentiation status described in the clinical manifest are from the clinical records and pathology reports and we apologise for any confusion caused. Addition of this information has now enabled us to compare DvP signature scores not just with differentiation status but also with tumour depth, tumour diameter, invasion status, Age and Sex (New Supplementary Figure 4d-f). Importantly tumour depth and diameter significantly correlate with progenitor like status whereas age and sex do not. We were only able to retrieve definitive invasion status data (beyond fat or perineural or lympho-vascular) from the clinical records of 7 of our samples. These had a tendency to be more progenitor like but this failed to reach statistical significance. Taken together our analyses indicate that progenitor like status correlates with tumour size, depth and pathological differentiation status further supporting our classification of disease progression and we hope the reviewer agrees that this now adds further to the significance of our findings.

2. Line 109: Extended Data Fig. 1e does show that normal skin and AK samples mostly cluster into class 1 but there are several AKs clustering well within Class 2 (cSCC and met) samples. Authors should comment on these – were these perhaps wrongly designated as AKs by histopath?

We thank the reviewer for highlighting this issue and we have now more specifically commented on these samples and added the following statements to our discussion, “We found that whilst lesions broadly cluster with clinicopathologic definitions many samples cluster atypically, notably including several AK samples which cluster with the majority of cSCC and metastasis samples. These may represent AKs with a high chance of progression to cSCC or pathologically misclassified samples” (Line 495-499). This further supports our conclusion that that cSCC disease progression is perhaps best classified as a disease continuum in which lesions fall on a spectrum from more differentiated to a progenitor-like state.

3. Line 123- 124 suggests that the genesets in figure 1a were derived from 1 (only?) published dataset but it isn't clear from the related Line 539 in figure 1 legend: “Consensus clustering identifies two classes of samples identified by RNAseq.” whether they are referring to data from their study or from the literature?? This needs to be clarified and the use of data from only one organotypic model justified further.

We apologise for the confusion and have amended the figure legend to read “ Consensus clustering of 110 human human samples profiled by RNAseq identifies two classes of

samples” (Lines 573-4). We utilised the progenitor, early and late differentiation gene signatures were defined by Paul A. Khavari's group (Lopez-Pajares, V., et al., A LncRNA-MAF:MAFB transcription factor network regulates epidermal differentiation. *Dev Cell*, 2015. 32(6): p. 693-706.) using organotypic models of epidermal differentiation in our study. This paper contained high quality RNAseq data which enabled us to generate the DvP signature score. This then enabled us to go on and perform the rest of our analyses and make the important observation that cSCC disease progression is best described as a disease continuum which then subsequently enables interrogation of the biological events that are coordinately regulated during disease progression. We feel that the quality of the Khavari lab data coupled with the ensuing enlightenment of our data set with input from single cell sequencing data provided in other high quality datasets (Ji et al., 2020, Takahashi et al., 2020) is sufficient justification for using this dataset.

4. Not surprising that “ normal skin and AK are significantly enriched for genes associated with late epidermal differentiation, whereas primary and metastatic SCC are significantly enriched for genes associated with skin progenitor cells (Fig. 1b).” and that cSCC “represents a continuum of epidermal de-differentiation” (Fig 1c). The DP score shows this continuum reasonably well for skin v AK v tumour separation but there seems to be quite a bit of intermingling of differentiation status, esp towards the right hand side which reduces the confidence in this “DP signature score”.

Whilst it is indeed not surprising that primary and metastatic SCC are significantly enriched for genes associated with skin progenitor cells we believe our study is the first one to comprehensively illustrate that this is indeed the case. The reviewer also raises an interesting point that as samples exhibit a more progenitor like state there is “quite a bit of intermingling of differentiation status”. Most of the samples in the final quartile of the DvP continuum are from samples classified by pathology report as poorly differentiated but there are several samples classified as MD/PD and one sample classified as WD/MD. This we believe highlights the difficulties pathologists face in classifying heterogenous cSCC tumours using H&E-stained sections of tumour biopsies. Rather than reducing confidence in the DP signature score as suggested by the reviewer we favour that this supports the use of an unbiased gene expression based approach to classify such complex tumours. This argument we believe is further supported by our findings shown in new Supplementary Figure 4 which illustrate that using a progenitor DvP or Quartile 1 vs Quartile 2 metagene analysis further separates human primary tumour samples by pathologist called differentiation status.

5. Extended data fig 2 legends needs more detail – eg are medians +/- what range shown here? The separated WD-MD (n = 2 samples) data set is not useful

We apologise for the omission to the figure legend and have now corrected this. We have also removed the WD-MD comparisons from our analyses and the revised figure is now presented as new Supplementary Figure 4.

6. Line 229 – extended data fig 4a - I dont see a clear separation between immune pathways and cell types esp for TKS and Tumor-KCs-diff

Our manuscript stated in line 229 “Significant changes in expression patterns of immune inhibitory and stimulatory pathways were also observed coincident with immune cell population changes (Extended Data Figure 4a, Supplementary Data File 5)”. We did not refer to the changes in keratinocyte populations at this point. To highlight the comparison between immune inhibitory and stimulatory pathways and immune cell types the revised manuscript now reads “Significant changes in expression patterns of immune inhibitory and stimulatory pathways were also observed coincident with immune cell population changes (Fig 3a, Supplementary Fig. 10a; Supplementary Data File 10)” and we hope this now makes this comparison clear. The reviewer does however raise an interesting point with regards to keratinocyte populations and immune modulatory gene expression and cell types but we do believe that comparison of Figure 3a and Figure 3d shows a clear coordinated change in immune cell populations and an increase in the preponderance of Tumour-Kc-Diff and TSK populations. Furthermore, in Figure 3c we highlight correlations of inhibitory and stimulatory immune genes with keratinocyte populations, in Figure 3d we clearly show a correlation with EDC gene expression and keratinocyte populations and in Figure 3e the EDC genes are shown to significantly correlate with immune cell types and gene expression modules.

7. Figure 2 legend: by “Heatmap shows 9 core co-expressed gene clusters ordered by DP signature score” do the authors actually mean by differentiation status as per the legend? Please provide more information regarding this statement: "Nine of these clusters exhibited significant correlation with the Differentiation-Progenitor score (Fig. 2a)", supplementary data etc.

We apologise for any confusion. The K-means clustering identified 15 clusters of co-ordinately expressed genes and 9 of these clusters exhibited significant correlation with the DvP signature score as illustrated in Figure 2a. The legend does not refer to differentiation status and the genes in the clusters and the expression levels of the genes in these clusters are shown in the heatmaps with the samples ordered by DvP signature score. We have amended the figure legend to now read “Heatmap shows gene expression levels of genes in 9 core co-expressed gene clusters with samples ordered by DvP signature score” and we hope this now clarifies the situation.

8. And related text (Lines 255-259): authors refer to cluster 3 as “enriched for pathways involved in formation of the cornified envelope, keratinization, neutrophil degranulation as well as antimicrobial peptides (Fig. 2a).” Should they be referring to cluster 2 not 3?? In any case, it looks like several other clusters either precede/are coincident with changes in immunomodulatory processes and/or other non-immuno processes. Anyway, it is not at all clear how the authors arrived at their conclusions in reference to Figure 2a,b.

We are grateful for the reviewer for pointing out our error in referring to cluster 3 instead of cluster 2 and we have now corrected this in the manuscript. We do believe that Figure 2a and b illustrate the coordinated changes in pathways and processes during disease progression and our analysis following on from this supports our conclusions that these represent important pathways and processes. We hope that our improved manuscript which contains several new analyses now also provides greater clarity.

Bioinformatics comments

1. Clarify whether the statements in the following two lines refer to data selected from the PCA plot (top 2k variable genes, Extended Data Figure 1)

Line 162: To decipher the molecular pathways and/or processes underpinning SCC progression we performed k-means clustering using the 2000 most significantly differentially expressed genes between clinical designations.

Line 687: Hierarchical k-means clustering was performed using normalised gene expression representing the 2000 most differentially expressed genes between the differentiated and progenitor-like states.

We apologise for the confusion surrounding the use of the 2000 most variable genes used as input for the PCA analysis and those used for the k-means clustering. The PCA analysis used the 2000 most variable genes as determined by median absolute deviation of the LogR normalised counts data. We have amended line 676 of the methods and materials to now read, "LogR normalised gene expression values were used for patient clustering. Hierarchical, PCA and tSNE analyses were implemented by the R packages, FactoMineR[56, factoextra[57] and Rtsne[58], respectively. The 2000 most variable genes as determined by median absolute deviation of the LogR normalised counts data were used for unsupervised clustering" (new lines 617-620). We have also updated the k-means clustering materials and methods section to now read "Hierarchical k-means clustering was performed using normalised gene expression representing the 2000 most differentially expressed (unique) genes between clinical designations i.e normal versus primary, normal versus AK, AK versus primary using $\text{padj} \leq 0.05$ & $\text{abs}(\log_2\text{FoldChange}) \geq \log_2(2)$ as the gene selection cutoff. Hierarchical k-means clustering was implemented by the `hkmeans` function in the `factoextra` R package[72]. The number of k clusters was estimated by "gap" statistic using the `cluster` R package[82]. The enrichment of a given k-means cluster in a patient sample was determined using `ssGSEA` as implemented by the `GSVA` R package[79]. Individual k-means signature scores were then correlated with the DvP score. k-means clusters were ranked according to their correlated enrichment scores and visualisation (Lines 632-641). We hope this now makes these issues clear.

2. Supp file 2; Please provide full GSEA result Tables and which annotation type was used (GO/Reactome) and list the query genes or number of genes (In the method section it only says; "using genes significantly and differentially expressed").

Excel entries shows Reactome, whereas tab name reads GO?

We apologise for these omissions and errors. The GSEA results are now shown in New Supplementary Data File 7 and now include complete GSEA results for Reactome and KEGG pathway enrichment analyses. The pathway annotations are clearly identifiable in the amended tables. Excel tabs have also been renamed to more accurately reflect the tables and the Supplementary Data File descriptor now also reads "GSEA using curated Reactome pathways showing significantly enriched molecular pathways and/or processes for each k-means cluster and GSEA using curated KEGG pathways showing significantly enriched molecular pathways and/or processes for each k-means cluster." We have also amended the methods section to now state "GSEA was performed using genes significantly and differentially expressed between the indicated clinical designations or genetic backgrounds. A cutoff of $\text{padj} \leq 0.05$ & $\text{abs}(\log_2\text{FoldChange}) \geq \log_2(2)$ was used to select genes for GSEA." (Lines 623-626).

Reviewer Figure 1. Heatmap of indicated genes previously implicated as being important in cSCC development and progression. Human samples profiled by RNASeq are arranged in order of DvP signature score. Clinical designations, immune status and differentiation status of the samples are also illustrated.

Reviewer Figure 2. Heatmap of gene expression of receptor tyrosine kinases in murine models of cSCC. Genotypes of tumours profiled by RNAseq are shown on the figure ((LPT, Deletion of *Tgfr2* coupled with Trp53loss/mutational activation coupled with additional deletion of *Notch1* (LNPT)), Deletion of *Tgfr2* coupled with knock in of *Kras*^{G12D} (LKT)).

REVIEWERS' COMMENTS

Reviewer #1 (Remarks to the Author):

The manuscript's authors entitled "Driver gene combinations dictate cutaneous squamous cell carcinoma disease continuum progression" have done an excellent job revising the manuscript and addressing the reviewers' comments. This reviewer is satisfied with the rebuttal and believes that the manuscript in its revised version is suitable for publication in Nature Communications.

Reviewer #2 (Remarks to the Author):

The authors have done a great job at responding to the reviewer queries. They have added significant new analyses that yielded interesting insights into metabolism, cell of origin, etc. While there is still no in situ confirmation or analysis, this study represents an excellent resource for the field.

Reviewer #3 (Remarks to the Author):

All my concerns have been addressed.

Reviewer #5 (Remarks to the Author):

General points:

In reading the manuscript and the reviewer comments with a focus on reviewer 4, there is no question that the reports adds to our current understanding of human SCC development. Its primary novelty remains the addition of more samples to a previously explored question and some limited new molecular insight.

The link to differentiation, while interesting, is neither novel nor unexpected, and has been expounded upon in various forms by prior papers Ji et al. (REF 20) and Chitsazzadeh et al. 2016 (REF 46), the latter of which has only been cited upon revision. Furthermore, while it is understood that there is little opportunity to link this back to any tissue / in-situ analysis, this largely prevents the authors from really taking this home and closing the circle in terms of mechanism. While the amount of work with the mouse models is impressive and described in detail with multiplexed IF as well, it is unclear what it ultimately adds. No AK-like lesions are assessed and histology is not reported so statements such as the LPK tumors resemble early lesions and LKT tumors resemble more SCC and the LPNT tumors represent intermediates has no tissue basis. Furthermore it is not clear that the order of these mutations as acquired in human disease is similar.

As far as the major conclusions in the paper are concerned, since the authors have brought Chitsazzadeh et al into the paper now, that the conclusions from human data from that paper are not discussed. The basic pattern of two major groups with AK overlapping across normal skin and SCC samples is replicated here and previously reported there. However, pathways of differentiation were discussed but not highlighted in that report.

Specific points:

1. The OncoPrint analysis including cBioportal data is welcome.
2. The authors have brought in more literature as asked and extended their analysis of the mouse data in particular, which is very welcome and significant. However, the Chitsazzadeh et al. 2016 paper also has human data which remains undiscussed. This paper, albeit utilizing a significantly smaller number of samples and more limited analysis, also reported cross species analysis of the

development of SCC and details the first next generation sequencing effort in this space.

Furthermore, conclusions such as “Our cross-species analysis indicates that driver gene combinations directly influence disease progression with progressive activation of ERK signalling providing a potential therapeutic target for cSCC.” contains no references. This conclusion was arrived at in the Chitsazzadeh et al. 2016 paper published in Nature Communications (Figure 5a) – “The key transcriptional drivers we identified in both pairwise signatures and the LME model are important early in the NS/CHR to AK/PAP transition and include E2F, ELK1 and NFY. Globally, we identified ERK signalling through ETS2 and ELK1, β -catenin signalling through TCF3 and LEF1, and possible differentiation pathways regulated by NFAT and AP1 as potentially important drivers of cuSCC development (Fig. 5).” This concept was also validated in vivo subsequently by Adelman et al 2016 <https://pubmed.ncbi.nlm.nih.gov/27293029/>

At the very least, recognition of these papers, though published 7 years ago, appears very much to have arrived at similar conclusions. Furthermore, data from this paper was included in the Chang and Hunter et al. (REF. 14) meta-analysis and should be mentioned up front in the introduction.

Overall however, the authors do only a partially acceptable job of really placing this prior work in the context of prior work.

3. The lack of corresponding histological analysis and response around that is ultimately problematic; of the claim is of misdiagnosis, this can be addressed. These patient samples must have had accompanying tissue to allow for diagnosis – perhaps central review by an appropriate pathologist would be warranted.

4. Along the lines of point 3 – what are the histologic features of the mouse models ? If the argument is to be made that LPT tumors represent early disease and LKT disease later disease, is that reflected in the histology of these tumors ?

5. Source of samples – normal skin is often listed with no site yet referred to has sun exposed. How is it known that it is sun exposed ?

6. Supplementary Figure 1 – is there overlap in pathways across the multiple pairwise comparisons ? This would be significant highlight if present and surprising if not present.

7. Supplementary Figure 7 – The term correlated might be preferred to coordinated as the latter might portend a mechanistic insight (e.g. specific transcription factors involved).

8. Supplementary Figure 13 – Are these putative drivers also highlighted in the Chang and Shain paper ?

9. The definition of immunosuppressed is not specified. If drug related, it would be ideal to know the drugs, context, and whether drug was being administered at the time lesions were harvested.

RESPONSE TO REVIEWERS' COMMENTS

Reviewer #1 (Remarks to the Author):

The manuscript's authors entitled "Driver gene combinations dictate cutaneous squamous cell carcinoma disease continuum progression" have done an excellent job revising the manuscript and addressing the reviewers' comments. This reviewer is satisfied with the rebuttal and believes that the manuscript in its revised version is suitable for publication in Nature Communications.

Reviewer #2 (Remarks to the Author):

The authors have done a great job at responding to the reviewer queries. They have added significant new analyses that yielded interesting insights into metabolism, cell of origin, etc. While there is still no in situ confirmation or analysis, this study represents an excellent resource for the field.

Reviewer #3 (Remarks to the Author):

All my concerns have been addressed.

Reviewer #5 (Remarks to the Author):

General points:

In reading the manuscript and the reviewer comments with a focus on reviewer 4, there is no question that the reports adds to our current understanding of human SCC development. Its primary novelty remains the addition of more samples to a previously explored question and some limited new molecular insight.

The link to differentiation, while interesting, is neither novel nor unexpected, and has been expounded upon in various forms by prior papers Ji et al. (REF 20) and Chitsazzadeh et al. 2016 (REF 46), the latter of which has only been cited upon revision. Furthermore, while it is understood that there is little opportunity to link this back to any tissue / in-situ analysis, this largely prevents the authors from really taking this home and closing the circle in terms of mechanism. While the amount of work with the mouse models is impressive and described in detail with multiplexed IF as well, it is unclear what it ultimately adds. No AK-like lesions are assessed and histology is not reported so statements such as the LPK tumors resemble early lesions and LKT tumors resemble

more SCC and the LPNT tumors represent intermediates has no tissue basis. Furthermore it is not clear that the order of these mutations as acquired in human disease is similar.

As far as the major conclusions in the paper are concerned, since the authors have brought Chitsazzadeh et al into the paper now, that the conclusions from human data from that paper are not discussed. The basic pattern of two major groups with AK overlapping across normal skin and SCC samples is replicated here and previously reported there. However, pathways of differentiation were discussed but not highlighted in that report.

Specific points:

1. The OncoPrint analysis including cBioportal data is welcome.
2. The authors have brought in more literature as asked and extended their analysis of the mouse data in particular, which is very welcome and significant. However, the Chitsazzadeh et al. 2016 paper also has human data which remains undiscussed. This paper, albeit utilizing a significantly smaller number of samples and more limited analysis, also reported cross species analysis of the development of SCC and details the first next generation sequencing effort in this space.

Furthermore, conclusions such as "Our cross-species analysis indicates that driver gene combinations directly influence disease progression with progressive activation of ERK signalling providing a potential therapeutic target for cSCC." contains no references. This conclusion was arrived at in the Chitsazzadeh et al. 2016 paper published in Nature Communications (Figure 5a) – "The key transcriptional drivers we identified in both pairwise signatures and the LME model are important early in the NS/CHR to AK/PAP transition and include E2F, ELK1 and NFY. Globally, we identified ERK signalling through ETS2 and ELK1, β -catenin signalling through TCF3 and LEF1, and possible differentiation pathways regulated by NFAT and AP1 as potentially important drivers of cuSCC development (Fig. 5)." This concept was also validated in in vivo subsequently by Adelman et al 2016 <https://pubmed.ncbi.nlm.nih.gov/27293029/>
At the very least, recognition of these papers, though published 7 years ago, appears very much to have arrived at similar conclusions. Furthermore, data from this paper was included in the Chang and Hunter et al. (REF. 14) meta-analysis and should be mentioned up front in the introduction.

Overall however, the authors do only a partially acceptable job of really placing this prior work in the context of prior work.

3. The lack of corresponding histological analysis and response around that is ultimately problematic; of the claim is of misdiagnosis, this can be addressed. These patient samples must have had accompanying tissue to allow for diagnosis – perhaps central review by an appropriate pathologist would be warranted.

4. Along the lines of point 3 – what are the histologic features of the mouse models ? If the argument is to be made that LPT tumors represent early disease and LKT disease later disease, is that reflected in the histology of these tumors ?

5. Source of samples – normal skin is often listed with no site yet referred to has sun exposed. How is it known that it is sun exposed ?

6. Supplementary Figure 1 – is there overlap in pathways across the multiple pairwise comparisons ? This would be significant highlight if present and surprising if not present.

7. Supplementary Figure 7 – The term correlated might be preferred to coordinated as the latter might portend a mechanistic insight (e.g. specific transcription factors involved).

8. Supplementary Figure 13 – Are these putative drivers also highlighted in the Chang and Shain paper ?

9. The definition of immunosuppressed is not specified. If drug related, it would be ideal to know the drugs, context, and whether drug was being administered at the time lesions were harvested.

Response to reviewers.

Reviewer #1: clinical expertise in actinic keratosis

Reviewer #2: expertise in skin cancer progression and evolution

Reviewer #3: expertise in melanoma mouse models

Reviewer #4: expertise in skin cancer sequencing

New Reviewer #5:

We are very grateful to the reviewers for their time spent reviewing our manuscript and for their mainly positive comments and constructive criticisms. We have taken on board their comments and revised our manuscript accordingly. Below, we have provided a point-by-point response to the reviewers' comments, with their comments in black print and our responses highlighted in blue. We believe that this has further improved our manuscript.

Reviewer #1 (Remarks to the Author):

The manuscript's authors entitled "Driver gene combinations dictate cutaneous squamous cell carcinoma disease continuum progression" have done an excellent job revising the manuscript and addressing the reviewers' comments. This reviewer is satisfied with the rebuttal and believes that the manuscript in its revised version is suitable for publication in Nature Communications.

We thank the reviewer for appreciating our revisions and indicating that our manuscript is now suitable for publication.

Reviewer #2 (Remarks to the Author):

The authors have done a great job at responding to the reviewer queries. They have added significant new analyses that yielded interesting insights into metabolism, cell of origin, etc. While there is still no in situ confirmation or analysis, this study represents an excellent resource for the field.

We thank the reviewer for appreciating our revisions and indicating that our study will provide an excellent resource.

Reviewer #3 (Remarks to the Author):

All my concerns have been addressed.

We are very pleased that all the reviewer's concerns were addressed by our previous resubmission.

Reviewer #5 (Remarks to the Author):

We would firstly like to express our gratitude to the reviewer for reviewing our manuscript at this stage of the process and for adding some additional important general and specific points for us to address. We believe this has further strengthened our manuscript.

General points:

In reading the manuscript and the reviewer comments with a focus on reviewer 4, there is no question that the reports adds to our current understanding of human SCC development. Its primary novelty remains the addition of more samples to a previously explored question and some limited new molecular insight.

We thank the reviewer for appreciating that our study adds to our understanding of cSCC.

The link to differentiation, while interesting, is neither novel nor unexpected, and has been expounded upon in various forms by prior papers Ji et al. (REF 20) and Chitsazzadeh et al. 2016 (REF 46), the latter of which has only been cited upon revision. Furthermore, while it is understood that there is little opportunity to link this back to any tissue / in-situ analysis, this largely prevents the authors from really taking this home and closing the circle in terms of mechanism. While the amount of work with the mouse models is impressive and described in detail with multiplexed IF as well, it is unclear what it ultimately adds. No AK-like lesions are assessed and histology is not reported so statements such as the LPK tumors resemble early lesions and LKT tumors resemble more SCC and the LPNT tumors represent intermediates has no tissue basis. Furthermore it is not clear that the order of these mutations as acquired in human disease is similar.

We thank the reviewer for being impressed with our mouse model work. We believe that this adds considerably to our study as it demonstrates experimentally that driver gene combinations are responsible for cSCC disease progression. The order of mutational acquisition in humans has not been studied in detail so far, however, our analysis illustrated here indicates that driver gene combinations correlate with DvP axis position and that when we recapitulate some of these combinations in the mouse, the murine tumours analysed from these mice display similar and DvP axis corresponding transcriptional profiles. We have now pathologically assessed the differentiation status of some of our murine tumours (New Supplementary Figure 15, new data sheet in Supplementary Data File 13, and mentioned and discussed this in our revised manuscript).

Most murine tumours are moderately differentiated yet transcriptional analysis indicates that these tumours represent a broad spectrum of the DvP axis which we believe further illustrates the power of transcriptional analyses.

As far as the major conclusions in the paper are concerned, since the authors have brought Chitsazzadeh et al into the paper now, that the conclusions from human data from that paper are not discussed. The basic pattern of two major groups with AK overlapping across normal skin and SCC samples is replicated here and previously reported there. However, pathways of differentiation were discussed but not highlighted in that report.

The data in Chitsazzadeh importantly indicated that human AK and cSCC samples both exhibit significant mutational heterogeneity and share mutations in common driver genes as also found in the other studies highlighted in the introduction, results section and discussion of our manuscript. We have now added the "mutational heterogeneity" statement to the introduction and the statement "Previous cross species studies integrating gene expression and genetic analysis have revealed potential mechanisms of cSCC disease progression" and added the Chitsazzadeh reference to the introduction in both places. We apologise for previously omitting this reference from the introduction.

Specific points:

1. The OncoPrint analysis including cBioportal data is welcome.

We thank the reviewer for appreciating this inclusion.

2. The authors have brought in more literature as asked and extended their analysis of the mouse data in particular, which is very welcome and significant. However, the Chitsazzadeh et al. 2016 paper also has human data which remains undiscussed. This paper, albeit utilizing a significantly smaller number of samples and more limited analysis, also reported cross species analysis of the development of SCC and details the first next generation sequencing effort in this space.

Furthermore, conclusions such as "Our cross-species analysis indicates that driver gene combinations directly influence disease progression with progressive activation of ERK signalling providing a potential therapeutic target for cSCC." contains no references. This conclusion was arrived at in the Chitsazzadeh et al. 2016 paper published in Nature Communications (Figure 5a) – "The key transcriptional drivers we identified in both pairwise signatures and the LME model are important early in the NS/CHR to AK/PAP transition and include E2F, ELK1 and NFY. Globally, we identified ERK signalling through ETS2 and ELK1, β -catenin signalling through TCF3 and LEF1, and possible differentiation

pathways regulated by NFAT and AP1 as potentially important drivers of cuSCC development (Fig. 5).” This concept was also validated in in vivo subsequently by Adelman et al 2016 <https://pubmed.ncbi.nlm.nih.gov/27293029/>

At the very least, recognition of these papers, though published 7 years ago, appears very much to have arrived at similar conclusions. Furthermore, data from this paper was included in the Chang and Hunter et al. (REF. 14) meta-analysis and should be mentioned up front in the introduction.

Overall however, the authors do only a partially acceptable job of really placing this prior work in the context of prior work.

The data in Chitsazzadeh importantly indicated that human AK and cSCC samples both exhibit significant mutational heterogeneity and share mutations in common driver genes as also found in the other studies highlighted in the introduction, results section and discussion of our manuscript. We have now added the Chitsazzadeh reference to the introduction and included additional statements as above. We apologise for previously omitting this reference from the introduction. We have now cited the Adelman paper and edited our discussion to highlight that both this paper and Chitsazzadeh et al revealed the potential of ERK signalling as a therapeutic target in cSCC. We apologise for the prior omission of Adelman et al and thank the reviewer for pointing this out.

3. The lack of corresponding histological analysis and response around that is ultimately problematic; of the claim is of misdiagnosis, this can be addressed. These patient samples must have had accompanying tissue to allow for diagnosis – perhaps central review by an appropriate pathologist would be warranted.

We have rereviewed the pathology notes for our human samples and find no evidence of misdiagnosis and so have removed this statement from the manuscript.

4. Along the lines of point 3 – what are the histologic features of the mouse models ? If the argument is to be made that LPT tumors represent early disease and LKT disease later disease, is that reflected in the histology of these tumors ?

We have now pathologically assessed the differentiation status of some of our murine tumours (New Supplementary Figure 15, new data sheet in Supplementary Data File 13, and mentioned and discussed this in our revised manuscript). Most murine tumours are moderately differentiated yet transcriptional analysis indicates that these tumours represent a broad spectrum of the DvP axis which we believe further illustrates the power of transcriptional analyses.

5. Source of samples – normal skin is often listed with no site yet referred to has sun exposed. How is it known that it is sun exposed ?

We thank the reviewer for pointing out this important omission. We have now added the site information to all of the samples in the clinical manifest tab of Supplementary Data

File 1. All of the normal skin samples are either perilesional or are taken from the sun exposed dorsum forearm.

6. Supplementary Figure 1 – is there overlap in pathways across the multiple pairwise comparisons ? This would be significant highlight if present and surprising if not present.

We thank the reviewer for asking us to comment on these potentially significant highlights and we have added the following statements to our results section “Importantly enrichment in cell cycle related processes were observed in both the normal to AK and normal to primary tumour comparisons indicating the hyperproliferative nature of both pre-malignant and malignant disease (Supplementary Fig 1b, d). Extracellular matrix processes were altered in normal-AK an AK-primary tumour transitions and cytokine, chemokine and IFN signalling pathways were enriched in primary tumour samples compared to both normal and AK samples indicating the potential importance of these events in primary tumour formation (Supplementary Fig. 1c,d).”

7. Supplementary Figure 7 – The term correlated might be preferred to coordinated as the latter might portend a mechanistic insight (e.g. specific transcription factors involved).

This is a good point and we have altered the figure title and legend and results section accordingly.

8. Supplementary Figure 13 – Are these putative drivers also highlighted in the Chang and Shain paper ?

The Chan and Shain paper highlighted NOTCH1, NOTCH2, HRAS and TP53 as putative drivers and our previous analyses and murine data included here have also implicated TGFBR1 and TGFBR2 as driver genes. Our GEMMs clearly illustrate that KRAS, TGFBR1, TGFBR2, TP53 and NOTCH1 can act as driver genes in murine cSCC.

9. The definition of immunosuppressed is not specified. If drug related, it would be ideal to know the drugs, context, and whether drug was being administered at the time lesions were harvested.

We thank the reviewer for pointing out this omission. One of our immunosuppressed patient samples was from a patient receiving mycophenylate mofetil, one had graft versus host disease and all others were from organ transplant patients. These details have been added to the clinical manifest tab in Supplementary Data file 1. Transplant patients receive variable immunosuppressant drug combinations both over time and between patients to prevent transplant rejection. We have previously highlighted the potential role of azathioprine in cSCC (Inman et al., 2018) and are embarking on further analyses to decipher the role of immune modulation in cSCC which is beyond the scope of the current study.